# Encystation stimuli sensing is mediated by adenylate cyclase AC2-dependent cAMP signaling in *Giardia*

Han-Wei Shih[1], Germain C. M. Alas[1] & Alexander R. Paredez ®[1] ✉

Protozoan parasites use cAMP signaling to precisely regulate the place and time of developmental differentiation, yet it is unclear how this signaling is initiated. Encystation of the intestinal parasite *Giardia lamblia* can be activated by multiple stimuli, which we hypothesize result in a common physiological change. We demonstrate that bile alters plasma membrane fluidity by reducing cholesterol-rich lipid microdomains, while alkaline pH enhances bile function. Through depletion of the cAMP producing enzyme Adenylate Cyclase 2 (AC2) and the use of a newly developed *Giardia*-specific cAMP sensor, we show that AC2 is necessary for encystation stimuli-induced cAMP upregulation and activation of downstream signaling. Conversely, over expression of AC2 or exogenous cAMP were sufficient to initiate encystation. Our findings indicate that encystation stimuli induce membrane reorganization, trigger AC2-dependent cAMP upregulation, and initiate encystation-specific gene expression, thereby advancing our understanding of a critical stage in the life cycle of a globally important parasite.

*Giardia lamblia* is a major cause of waterborne diarrheal disease in humans and other mammals throughout the world. This protozoan parasite exhibits a biphasic life cycle with a replicative trophozoite stage and an environmentally resistant cyst stage, which is the infective form for transmission. After oral ingestion of contaminated water or food, cysts release vegetative trophozoites that colonize the small intestine. In the small intestine, bile, alkaline pH, and cholesterol starvation are believed to play important roles in stimulating encystation and may provide positional cues to the parasite[1]. These stimuli induce the master transcription factor MYB2, which activates the expression of cyst wall proteins (CWPs) and enzymes involved in the production of β−1,3-N-acetylgalactosamine (GalNAc) homopolymers[2,3]. Together GalNac homopolymers and CWPs are used to construct the cyst wall, which is essential for survival outside the host and passage through the acidic environment of the stomach.

cAMP is a ubiquitous second messenger that regulates numerous metabolic and cellular processes in eukaryotic cells and is important for regulating differentiation in several parasites and *Dictyostelium discoideum*[4–6]. Encystation stimuli have been demonstrated to raise intracellular cyclic nucleotide adenosine 3′,5′-monophosphate (cAMP)

and re-localize the cAMP-regulated Protein Kinase A catalytic subunit (PKAc) after the induction of encystation[7,8]. PKA and calcium signaling, which is linked to cAMP regulation in other eukaryotes, have roles in regulating excystation indicating that cAMP and PKA function in both of *Giardia's* differentiation events[9,10]. The role of cAMP and downstream effector proteins in *Giardia* is poorly understood. While upregulated cAMP hints that cAMP plays a role in encystation activation, its precise role in encystation has remained untested.

Defining the molecular mechanism that regulates encystation is important because this process is essential for transmission to humans and other hosts and is overall poorly understood across the diversity of eukaryotes that form resting cysts[11]. *Giardia* is an excellent model for studying encystation since differentiation is readily induced in vitro using various encystation media that alter cholesterol availability and pH[12–14]. Additionally, *Giardia* has a compact genome containing a reduced subset of cAMP signaling machinery with minimal redundancy. These include two genes encoding the enzymes that synthesize cyclic nucleotides, transmembrane adenylate cyclases (AC1 and AC2). AC1 can synthesize cAMP and cGMP, while AC2 was predicted to only synthesize cAMP[7]. Only a single cAMP degrading enzyme

[1]Department of Biology, University of Washington, Seattle, WA 98195, USA. ✉e-mail: aparedez@uw.edu

phosphodiesterase (PDE) and a single cAMP effector protein cAMP-regulated kinase PKAc have been identified in *Giardia*[15].

In model eukaryotes cAMP signaling is initiated by G-protein coupled receptors (GPCRs, Fig. 1a). However, GPCRs and heterotrimeric G-proteins associated with stimulating adenylate cyclases are absent from Excavate and Alveolate parasites (Fig. 1b), suggesting that these parasites rely on unique molecular mechanism to regulate cAMP levels[16]. The relatively small number of *Giardia* cAMP components facilitated our experimental manipulation of key players through chemical and genetic means. Here, we show that plasma membrane cholesterol depletion elicits an AC2-dependent elevation of intracellular cAMP and the dissociation of PKA regulatory and catalytic subunits, thereby activating PKAc and encystation signal transduction.

## Results

### Altered plasma membrane fluidity is the key trigger for inducing encystation

In *Giardia*, cholesterol depletion and alkaline pH are two crucial stimuli to trigger encystation[12]. Cholesterol depletion can be achieved by growing cells in lipoprotein deficient medium or using the biological amphiphile bile to sequester cholesterol[12–14]. An alkaline pH of 7.8 enhances encystation rates for both bile-induced and lipoprotein-deficient serum methods[12,13]. To define the roles of bile and alkaline pH for the initiation of encystation, we examined whether elevated bile or alkaline pH were sufficient to trigger the production of an encystation marker, CWP1, endogenously tagged with the NanoLuc reporter. Alkaline pH itself did not upregulate CWP1-NLuc, but 0.25 mg/mL porcine bile was sufficient to remove membrane cholesterol as indicated by Cholera toxin subunit B, Alexa Fluor™ 594 conjugate (CTXB) staining[17] (supplemental Fig. 1a) and upregulate CWP1-NLuc (Fig. 2a). Bile treatment reduces the ordering (increases fluidity) of plasma membranes from mammalian cells and a previous study suggested that *Giardia* cholesterol deprivation could be altering plasma membrane cholesterol content and membrane fluidity[12,18]. We therefore tested if altered membrane fluidity changes the membrane composition and stimulates encystation by using the membrane cholesterol depletion agent methyl-β-cyclodextrin (MβCD). MβCD treatment resulted in cholesterol depletion (supplemental Fig. 1a) and led to a two-fold increase in CWP1 levels (Fig. 2a), indicating that altered plasma membrane fluidity is a major stimulation for the initiation of encystation. Next, our results indicated that exogenous cholesterol complements the membrane cholesterol depletion and the CWP1-NLuc expression from both MβCD treatment (supplemental Fig. 1b, d) and encystation medium

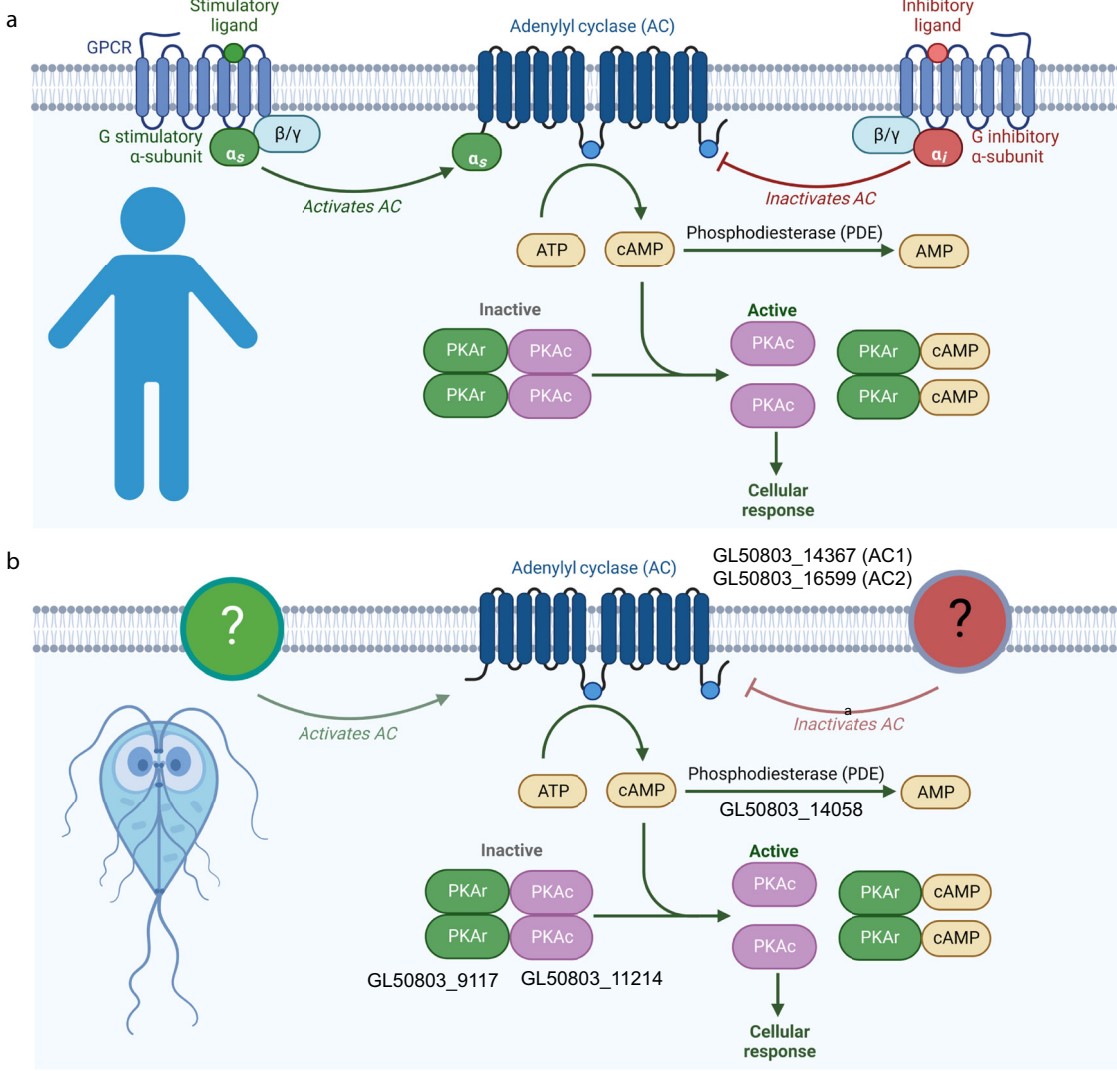

**Fig. 1 | cAMP signaling in humans versus *Giardia*.** Diagram depicts canonical cAMP signaling in humans (**a**) vs cAMP signaling in *Giardia* (**b**). Note that *Giardia* lacks G-protein coupled receptors and heterotrimeric G-proteins canonically linked to adenylate cyclase regulation. AC1 (GL50803_14367), AC2 (GL50803_16599), PDE (GL50803_14058), PKAr (Gl50803_9117), and PKAc (GL50803_11214). Created with BioRender.com.

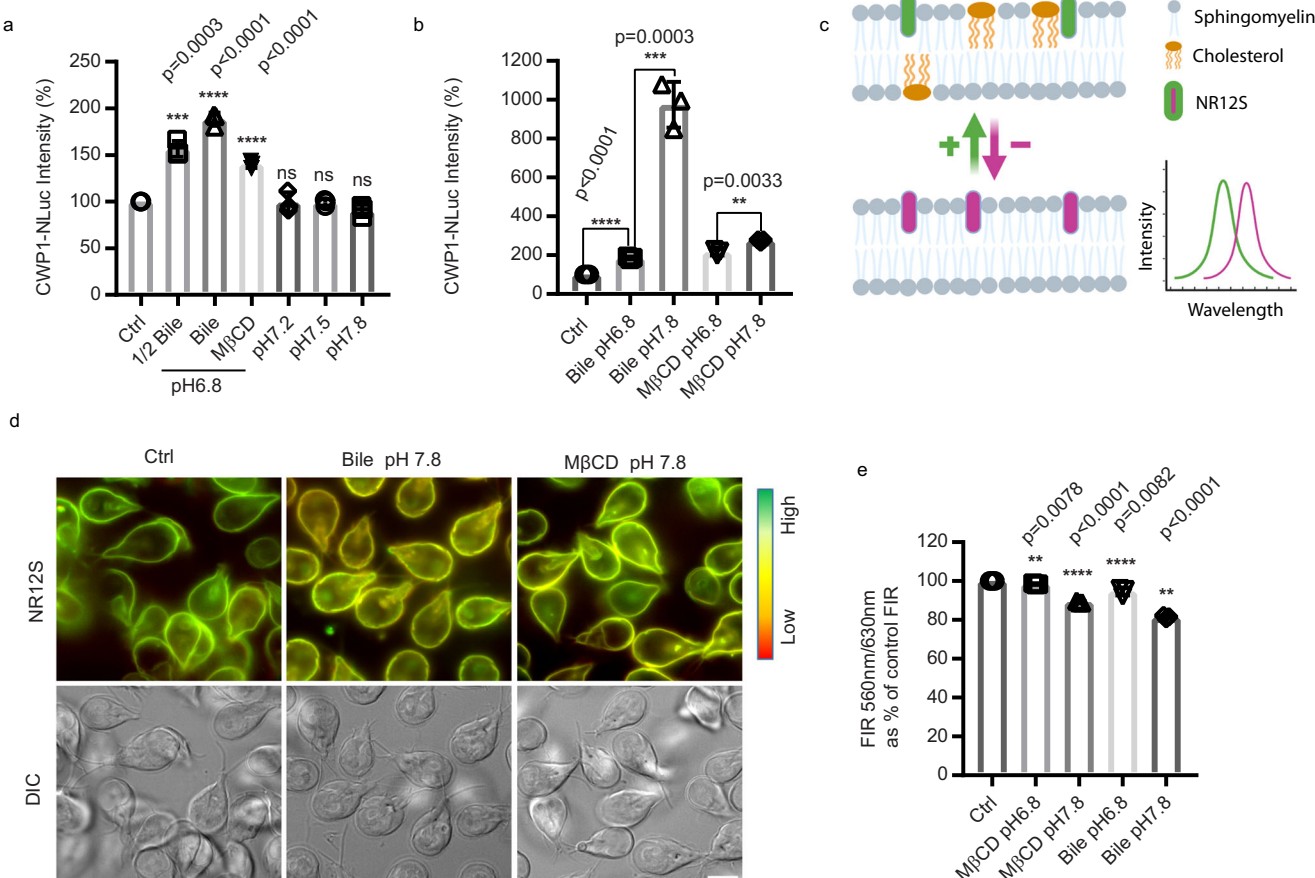

**Fig. 2 | Bile alters plasma membrane fluidity. a, b** Quantification of endogenously tagged CWP1-NLuc intensity after 4 h exposure to different encystation stimuli, including (**a**) 0.125 mg/mL porcine bile pH 6.8 (1/2 Bile) and 0.25 mg/mL porcine bile pH 6.8 (Bile), 50 μM MβCD pH 6.8, pH 7.2, pH 7.5, and pH 7.8, (**b**) the combination of 0.25 mg/ml porcine bile or 50 μM MβCD at either pH 6.8 or pH 7.8. CWP1-NLuc intensity is normalized to control. Data are presented as mean ± s.e.m from a minimum of three independent experiments. **c–e** Parasites were incubated with either porcine bile or MβCD at pH 6.8 or pH 7.8 for 1 h and incubated with 0.05 μM NR12S for 30 min to measure plasma membrane fluidity. **c** A schematic of NR12S imaging principle on the plasma membrane. Upper panel indicates lipid bilayer with cholesterol (yellow). Lower panel indicates the lipid bilayer after cholesterol depletion, red-shifted emission indicates less ordered lipids. Green arrow and magenta arrow represent increased or decreased membrane cholesterol and lipid order respectively. After cholesterol depletion, the fluorescence intensity of NR12 shifts from blue-green emission wavelength (560 nm) to red (630 nm). Created with BioRender.com. **d** Representative NR12S staining in live cells. **e** Quantification of ratiometric NR12S fluorescent intensity of parasites incubated with 0.25 mg/mL porcine bile or 50 μM MβCD at pH 6.8 or 7.8 for 1 h using plate reader with excitation 520 nm, emission 560 nm/630 nm; green indicates highly ordered membranes with high cholesterol and red indicates less ordered membranes with reduced cholesterol. Reduced fluorescence intensity ratio (FIR) 560 nm/630 nm indicates decreased plasma membrane cholesterol. Data are from three independent experiments. P values were calculated with two-tailed *t* tests. Scale bar, 5 μm.

(porcine bile) treatment (supplemental Fig. 1e). Furthermore, parasites treated with MβCD and bile incubation at pH 7.8 increased the level of CWP1-NLuc by 1.2 and 5-fold respectively (Fig. 2b), suggesting alkaline pH is important to enhance the induction of encystation but not initiate the response. This result, however, does not exclude the possibility that pH is being sensed by additional mechanisms.

To monitor changes in plasma membrane lipid ordering (membrane fluidity), we performed ratiometric measurements of the solvatochromic fluorescent dye Nile Red 12S (NR12S) via live imaging and plate reader assays. NR12S exclusively binds to the outer leaflet of the plasma membrane through its amphiphilic anchor group and its emission spectra correlates with plasma membrane sphingomyelin and cholesterol content. In ordered lipids (sphingomyelin and cholesterol) emission is blue shifted whereas the loss of cholesterol and saturated phospholipids results in reduced membrane ordering and red-shifted emission (Fig. 2c)[19,20]. Upon cholesterol depletion with MβCD lipid order is decreased and NR12s emission is red shift (Fig. 2d). We found that parasites incubated with MβCD for 1 h displayed a 2% reduction of the 560 nm/630 nm Fluorescence Intensity Ratio (FIR) at pH 6.8, and bile treatment showed 5% reduction at pH 6.8 (Fig. 2e).

MβCD and porcine bile showed 10% and 20% reductions at pH 7.8 respectively. Interestingly, exogenous cholesterol could complement the membrane cholesterol depletion from the bile treatment (supplemental Fig. 1f, g). Taken together, the results are consistent with bile-induced membrane cholesterol depletion being a key factor for the initiation of encystation in *Giardia*.

## Inducers of membrane cholesterol depletion trigger cAMP signaling

In *Giardia*, intracellular cAMP levels peak 1 h post induction of encystation[7]. We questioned whether and at what point membrane cholesterol depletion elicits upregulation of intracellular cAMP and activation of downstream cAMP signaling. PKAc is the only conserved cAMP effector protein in the *Giardia* genome and it has been implicated in the regulation of encystation[8]. Protein Kinase A regulator (PKAr) and catalytic (PKAc) subunits form an inactive signaling complex. Upon cAMP binding to PKAr, PKAc is allosterically released as an active kinase. To monitor this in live cells we generated a *Giardia* specific cAMP sensor that reports the presence of biologically relevant cAMP levels by measuring PKAr-PKAc interaction. Our strategy was to

fuse components of a split NanoLuc system (NanoBit)[21] to *Giardia* PKAr and PKAc, which would result in luminescence in the absence of cAMP and decreased luminescence when cAMP is bound by PKAr and PKAc is released to propagate signaling. A similar approach was used to generate Glosensor, a popular cAMP reporter that has split firefly luciferase fused to a cAMP binding domain from PKAr[22]. Glosensor did not work in *Giardia* perhaps because the human type II-beta regulatory subunit of PKA used in Glosensor has a binding affinity and range of response that is incompatible with cAMP levels found in *Giardia*.

To verify that a PKA-based sensor would be tractable in *Giardia*, we first examined expression levels of PKAr (GL50803_9117) and PKAc (GL50803_11214). A previous study found that the level of PKAr decreased in response to encystation stimuli[8], but this study did not include a loading control to accurately quantify PKAr levels. Using PKAr and PKAc cell lines that were endogenously tagged with Nano-Luc-3HA, we found no significant change in the level of the proteins at intervals up to 24 h post induction of encystation (Fig. 3a and Supplemental Fig. 2a, b). *Giardia* PKAr interacts with PKAc and exogenous membrane permeable cAMP enhances the dissociation of PKAr and PKAc[8]. In response to encystation stimuli, the localization of PKAc is

altered[8]. To examine the co-localization of PKAr and PKAc, the subunits were endogenously tagged with mNeonGreen (mNG) and the Haloalkane dehalogenase (Halo) tag respectively. Our results indicate co-localization throughout the cytoplasm and association with flagellar axonemes. After 2–4 h post induction of encystation co-localization at the anterior flagella but not the caudal flagella and cytosol was reduced (Fig. 3b). To quantify the PKAr-PKAc interaction and the activation of PKAc post induction of encystation, we generated a new *Giardia* specific cAMP signaling sensor, *Gl*PKA-NBit.

NanoBit is a structural complementation reporter system that has been used to monitor dynamic protein-protein interactions in a variety of cell types[21]. For the study of PKAr-PKAc interaction, the complimentary peptide Small BiT (SmBiT) was fused with PKAc driven by its native promoter, and the Large BiT (LgBiT) was fused to PKAr driven by its native promoter (Fig. 3c and Supplementary Fig. 2c). The control cell line expresses the PKAc-SmBiT and the LgBiT fragment driven by the PKAr native promoter (*p*PKAr) not fused to any protein (Supplementary Fig. 3c). Our data shows that only PKAr-LgBiT co-expressed with PKAc-SmBit (*Gl*PKA-Nbit) could produce significant luminescence after incubation with the NanoLuc substrate NanoGlo (Supplementary

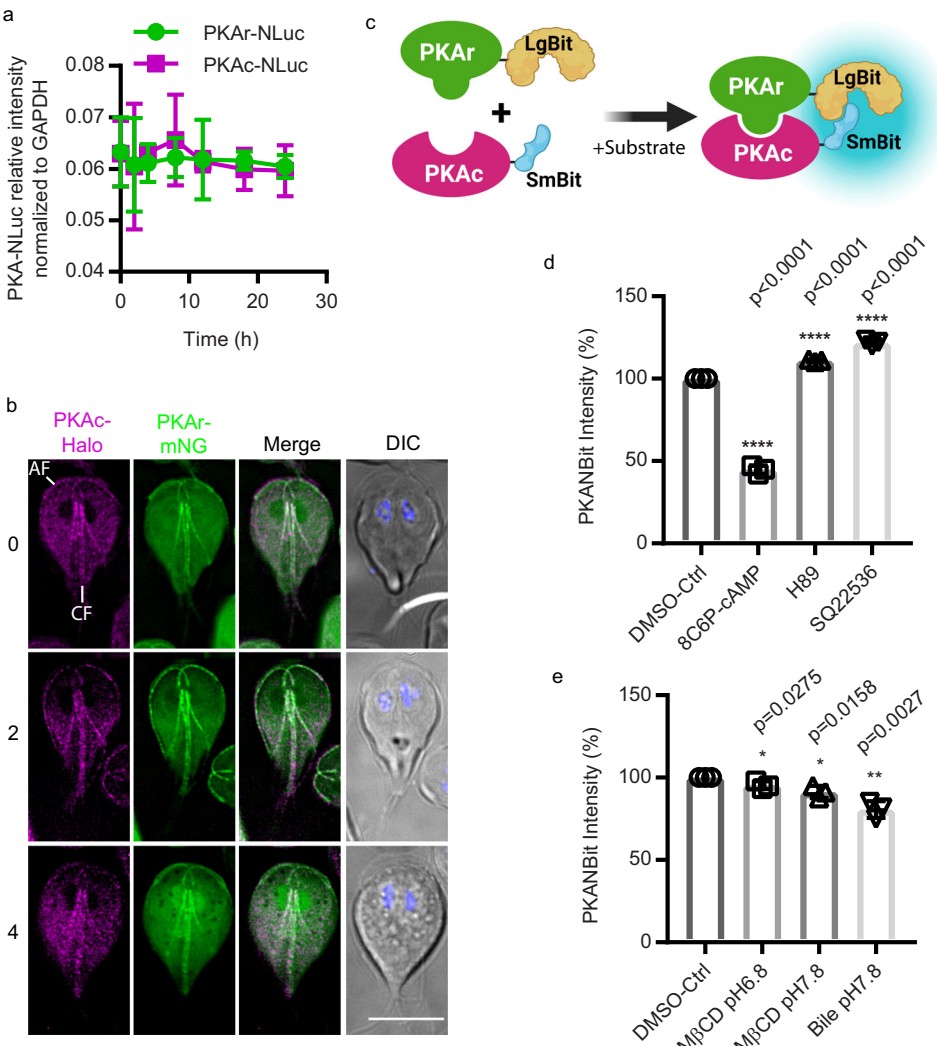

**Fig. 3 | Design of *Giardia* cAMP sensor- *Gl*PKA-NBit. a** Relative expression levels of PKAr-NLuc and PKAc-NLuc after 0, 2, 4, 8, 12, 16, and 24 h exposure to encystation medium. Expression level is normalized to GAPDH::NLuc intensity. **b** Localization of PKAr-Halo and PKAc-mNeonGreen (mNG) after 0, 2, and 4 h exposure to encystation stimuli. Anterior and caudal flagella are marked AF and CF respectively. Scale bar, 5 µm. **c** A schematic of PKA-NanoBit (*Gl*PKAr-LgBit and

*Gl*PKAc-SmBit). Created with BioRender.com. **d, e** Quantification of PKA-NBit intensity after 1 h exposure to (**d**) 50 µM 8C6P-cAMP, 10 µM H89, and 10 µM SQ22536, (**e**) bile or MβCD. **d, e** are normalized to control. All data are mean ± s.d. from three independent experiments. *P* values were calculated with two-tailed *t*-tests.

Fig. 2d). Bright luminescence indicated robust interaction between PKAr and PKAc which is consistent with prior protein-protein interaction studies[8]. To assess whether *Gl*PKA-Nbit responds to cAMP, *Gl*PKA-Nbit cell lines were incubated with a cAMP analog or inhibitors of cAMP signaling. In response to the membrane permeable and cleavage-resistant cAMP analog 8C6P-cAMP, the luminescent intensity of *Gl*PKA-Nbit decreased indicating the expected dissociation of PKAr and PKAc (Fig. 3d). The PKAc inhibitor H89[23] prevented the dissociation of PKAr and PKAc (Fig. 3d). More importantly, 9-(tetrahydrofuran-2-yl)−9h-purin-6-amine (SQ22536), a previously reported adenylate cyclase inhibitor[24], enhanced association of PKAr and PKAc, consistent with a decrease of intracellular cAMP (Fig. 3d). Together, these results indicate that the dissociation of *Gl*PKA-Nbit is cAMP-dependent.

We next used this reporter to determine whether cholesterol depletion agents could elicit the dissociation of *Gl*PKA-Nbit. *Gl*PKA-Nbit intensity decreased after a 1 h exposure to MβCD and bile, indicating membrane cholesterol depletion agents elicit an increase of intracellular cAMP that causes the dissociation of *Gl*PKA-Nbit and activation of PKAc (Fig. 2e). To understand the temporal dynamics of intracellular cAMP during encystation, parasites were exposed to high bile encystation medium and then examined at different time points. We found that the intensity of *Gl*PKA-NBit decreased by 30 min, consistent with encystation stimuli causing a rise in intercellular cAMP and disassociation of PKAr and PKAc (Supplementary Fig. 3e). Taken together, our newly developed signaling sensor *Gl*PKA-NBit confirms that altered membrane fluidity triggers elevation of intracellular cAMP and activation of PKAc.

### Exogenously supplied cAMP upregulates encystation rates

*Gl*PKA-NBit indicated that PKAc is activated by encystation, we questioned whether exogenously supplied cAMP could upregulate encystation[8]. Measurements of cAMP levels with ELISA assays confirmed that intracellular cAMP is upregulated by 30 min and peaks at 1 h post induction of encystation (Supplementary Fig. 3a). To better understand the role of cAMP in encystation, trophozoites were briefly incubated with three different membrane permeable and cleavage resistant cAMP analogs for 1 h before exposure to encystation medium. The cAMP analogs were washed out with pre-encystation medium and the cells were exposed to encystation medium without cAMP analogs. We used CWP1 as an encystation marker in western blots and immunofluorescence assays (IFA). Pulsed treatments of exogenous 8Br-cAMP, DB-cAMP, and 8C6P-cAMP resulted in 1.2-fold, 1.5-fold, and 2-fold increases of CWP1 at 4 h post induction of encystation respectively (Fig. 4a, b and Supplementary Fig. 3b, c). Further IFA experiments showed that CWP1 positive cells increased by 2-fold at 24 h post induction of encystation after a 1 h pretreatment of exogenous 8C6P-cAMP (Fig. 4c−e), thereby increasing total cyst production and cyst viability (Supplementary Fig. 3d, e). These results indicate that cAMP enhances encystation in response to encystation stimuli.

### AC2 is necessary for the initiation of cAMP-dependent encystation

To determine which adenylate cyclase, AC1 or AC2, is responsible for upregulated cAMP production at the initiation of encystation, we began by examining expression and subcellular localization of AC1 and AC2 endogenously tagged with mNeonGreen (mNG). Both proteins have predicted transmembrane domains and since plasma membrane cholesterol depletion upregulates cAMP levels, we anticipated that the adenylate cyclase(s) involved in the encystation response would be localized to the plasma membrane. Both N and C-terminally tagged AC1 localized to cytosolic puncta at 8 h and 16 h post induction of encystation (Fig. 5a and Supplementary Fig. 4a). Expression analysis confirmed that AC1 is specifically expressed in the mid-late stages of encystation (Supplementary Fig. 4b). AC2-mNG was expressed at all timepoints

examined and was predominantly localized to the plasma membrane (PM) (Fig. 5b) and the endoplasmic reticulum (ER) (Supplementary Fig. 5a). Interestingly, we also observed that AC2-mNG partially co-localizes with CTXB, indicating AC2 associates with lipid raft domains consistent with lipid raft proteomics[25] (supplemental Fig. 1c). At the cyst stage AC2 was found in an uncharacterized internal compartment that did not co-localize with ER or ESV markers (Supplementary Fig. 5b).

The presence of AC2 but not AC1 at the plasma membrane (PM) before encystation was induced suggested that AC2 was likely important for upregulated cAMP in response to altered membrane fluidity. To determine whether AC2 is required for the biosynthesis of cAMP during encystation, we depleted AC2 using CRISPRi-mediated knockdown[26] and measured cAMP with an ELISA assay 1 h after inducing encystation (Supplementary Fig. 6a). AC2-g4159 knockdown had lower intracellular cAMP than the dCas9 control (Fig. 5c). In agreement with lower cAMP production, the *Gl*PKA-NBit sensor indicated that bile-induced intracellular cAMP elevation was abolished in the AC2-g4159 knockdown line (Fig. 5d). These results indicate that AC2 is required for the dissociation of PKAr and PKAc (Fig. 5d), which confirms that AC2 is necessary for the synthesis of intracellular cAMP in response to encystation stimuli.

We next questioned whether the reduction in cAMP and PKAc activation would alter the encystation response. Since Myb2, has been shown to regulate CWP1, CWP2, CWP3, and GalNAc synthesis enzymes and is necessary for cyst formation[27], we used this important protein as a readout for encystation. We transformed the dCas9-Ctrl and AC2-g4159 knockdown constructs into MYB2-NLuc and CWP1-NLuc cell lines respectively so that their expression could be monitored. We found that AC2 knockdown reduced MYB2-Nluc and CWP1-NLuc levels at 4 and 8 h post induction of encystation compared to the control, indicating that AC2 is an upstream regulator of MYB2 and CWP1. Consistent with CRISPRi-based AC2 knockdown, morpholino-mediated knockdown of AC2 also reduced CWP1 at 4 h post induction of encystation (Fig. 5e−g and Supplementary 6b, c). We next tested if AC1 plays a role in the later stages of encystation. We transformed the dCas9-Ctrl and AC1 guide RNAs and found that AC1 is critical for the CWP1 production at 24 h post exposure to encystation (Supplementary Fig. 4c−e). However, overexpression of AC1-mNG does not activate CWP1 expression, suggesting AC1 is not necessary for the initiation of encystation but plays a role in later stages of encystation (Supplementary Fig. 4f−h). In summary, we determined that AC2 is required for the observed encystation-induced intracellular cAMP elevation which leads to upregulation of MYB2 and CWP1.

### AC2 is necessary for cyst production

*Giardia*'s cyst wall is composed of CWPs 1–3, High Cysteine Non-variant Cyst protein (HCNCp), and GalNAc fibrils that are only produced during encystation[28,29]. We asked whether AC2 additionally regulates CWP2, CWP3, HCNCp, and the five GalNAc biosynthesis enzymes G6PI-B, GNPNAT, UAP, UAE, PGM3). We transformed AC2-g4159 into cell lines with each of these components tagged with NanoLuc. In AC2 knockdown cell lines, all three CWPs, HCNCp, and four of the five GalNAc biosynthesis enzymes were downregulated, indicating that their levels are regulated through cAMP signaling (Fig. 6a−h). However, there was some variability in the level of reduction that suggests factors in addition to cAMP signaling have a role in their regulation.

The expression and localization of CWP1-3 are well characterized; however, HCNCp has only been localized during late encystation and the localization of GalNAc biosynthesis enzymes has never been reported. We endogenously tagged these genes with Halo and NLuc to study their localization and expression. Our results show that HCNCp co-localizes with CWP1 at ESVs by 8 h post-induction of encystation (Supplementary Fig. 7a, b). G6PI-B, GNPNAT, UAP, UAE, and PGM3 are highly upregulated by 16 h post induction of encystation and the

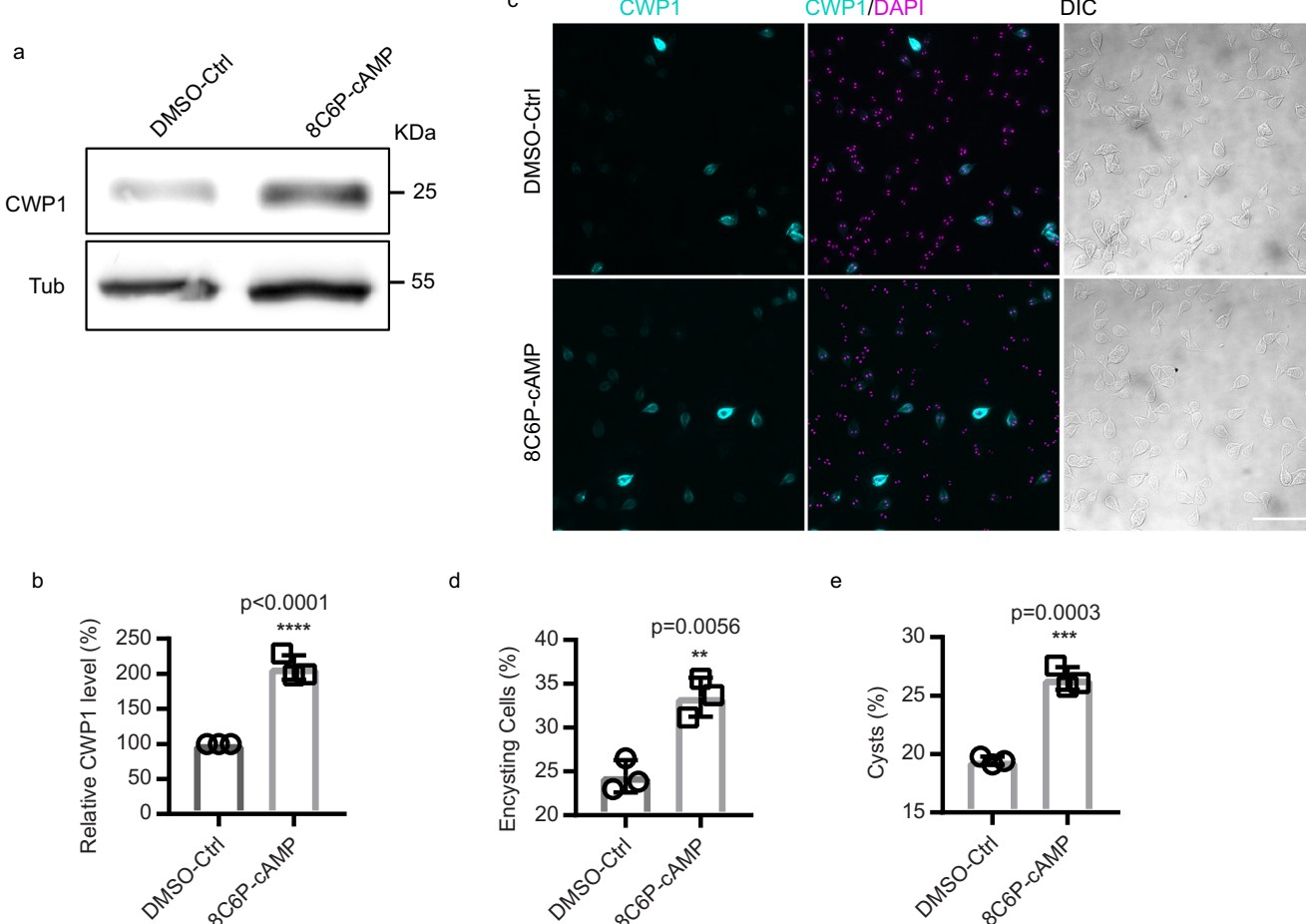

**Fig. 4 | Membrane permeable cAMP enhances encystation. a** Western blot of CWP1 and tubulin from control and 50 µM 8C6P-cAMP pretreatment. Wild type parasites were pretreated with 8C6P-cAMP for 1 h, washed with pre-encystation medium, and exposed to encystation medium for 4 h. **b** Quantification of (**a**), the expression level of CWP1 is normalized using tubulin as a loading control. **c** Representative images of encysting cells after 1 h pretreatment of 8C6P-cAMP

followed by 24 h exposure to encystation medium, and (**d**) quantification (total cells counted for DMSO-Ctrl *n* = 1002, and 8C6P-cAMP *n* = 997). Scale bar, 50 µm. **e** Quantification of mature cysts at 48 h post induction of encystation from parasite with or without 1 h of 8C6P-cAMP pretreatment. Cyst counts were performed by hemocytometer (cysts counted for DMSO-Ctrl *n* = 1730, and 8C6P-cAMP *n* = 2338). Data are mean ± s.d. from three independent experiments.

proteins were found to be cytoplasmic (Supplementary Fig. 7c–l), which agrees with previous fractionation studies[30]. We also tested if AC2 knockdown changes the expression of PDE and AC1. Our data shows that AC2 knockdown only reduces PDE expression but not AC1 expression (Supplementary Fig. 7m, n).

We next asked whether AC2 knockdown would alter the encystation rate and formation of mature cysts. In comparison to the dCas9 control, AC2 knockdown resulted in a 68% reduction in encysting cells, as monitored by CWP1 expression (Fig. 6i, j). Notably, exogenous membrane permeable 8C6P-cAMP rescued AC2 knockdown and increased the proportion of encysting cells 46% above the control (Supplementary 6d, e). AC2 knockdown reduced water-resistant cyst formation by 99%, indicating that AC2 is required for cyst formation (Fig. 6k). For the few cysts that formed there was a slight increase in viability (Supplementary Fig. 6f, g). This result indicated that AC2-dependent cAMP signaling is required for both initiation of encystation and cyst maturation.

As an orthogonal approach to disrupting AC2-mediated signaling, we tested whether adenylate cyclase inhibitor SQ22536 altered expression of CWP1 and the percentage of cells that encyst. We found that a 1 h pretreatment of 10 µM SQ22536 decreased CWP1 levels by 25% at 4 h post induction of encystation (Supplementary Fig. 8a, b), significantly decreased the level of encysting cells 24 h post induction

of encystation (Supplementary Fig. 8c, d), significantly decreased mature water-resistant cyst formation 48 h post induction of encystation (Supplementary Fig. 8e), and modestly increased the viability of the formed cysts (Supplementary Fig. 8f, g).

## AC2 is sufficient to initiate differentiation in the absence of encystation stimuli

We asked whether an exogenous membrane-permeable cAMP analog or AC2 overexpression would be sufficient to initiate differentiation in the absence of encystation stimuli. We found that membrane permeable 8C6P-cAMP triggered encystation as indicated by 27.9% CWP1 positive cells 24 h post treatment versus 0.54% in the control. Notably, these encysting cells did not differentiate into mature cysts (Fig. 7a, b), suggesting an additional signaling pathway may modulate encystation.

To complement exogenous cAMP additions, we generated an AC2 overexpression construct driven by the α tubulin promoter (pαTub::AC2-mNG) (Supplementary Fig. 9a). Overexpression driven by the αTub promoter increased intracellular cAMP as measured by a cAMP ELISA assay (Fig. 7c). Compared pαTub::mNG, AC2 over expression increased CWP1 levels by more than 9 fold in the absence of encystation stimulation (Fig. 7d, e), and immunofluorescence analysis indicated that 8.75% of pαTub::AC2-mNG cells were CWP1 positive compared to 0.25% of the control cells (Fig. 7f, g). We further

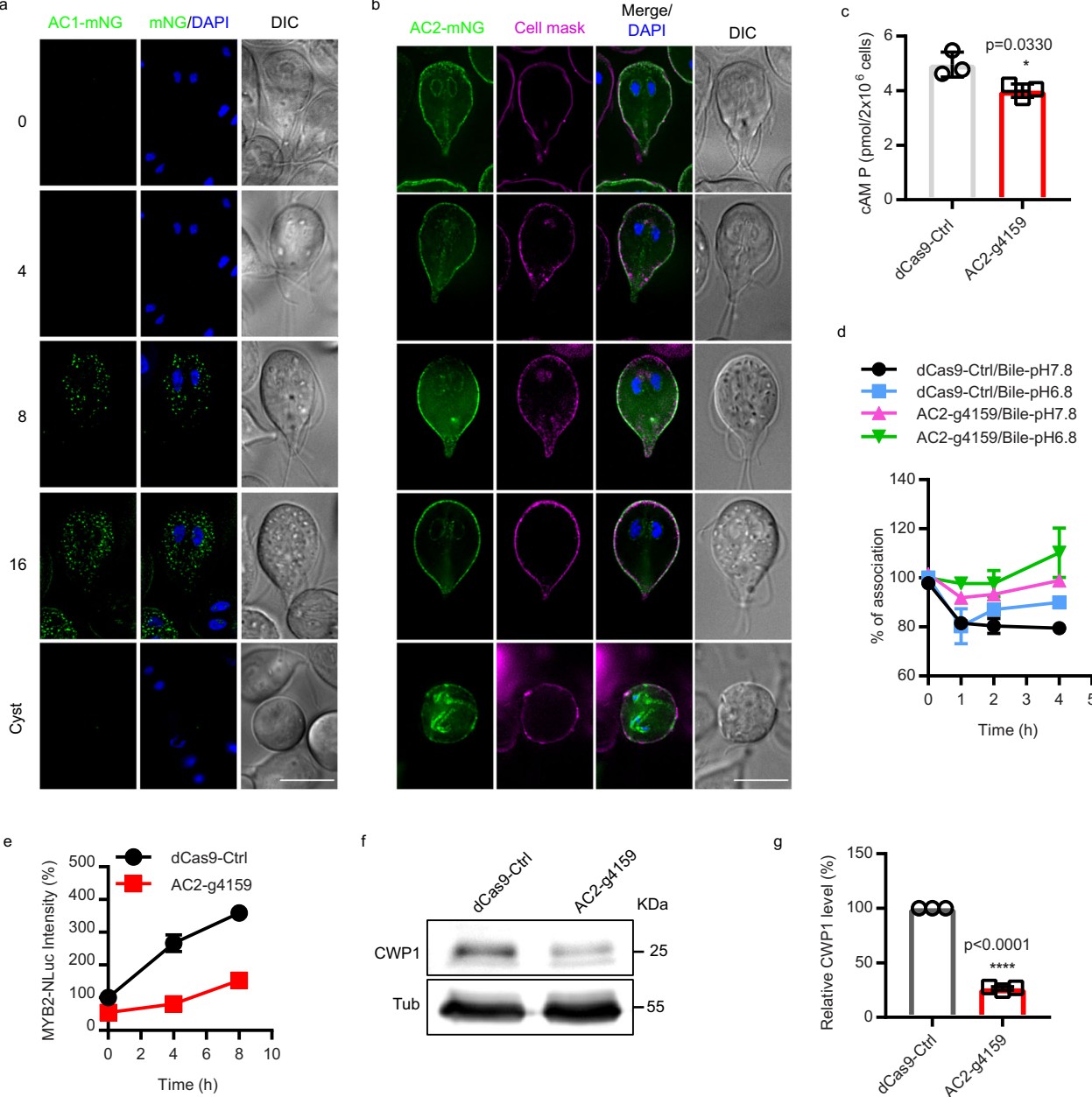

**Fig. 5 | gAC2 is essential for encystation-induced intracellular cAMP elevation.**
**a**, **b** Localization of AC1-mNG (**a**) and AC2-mNG (**b**) at 0, 4, 8, 16, and 24 h post exposure to encystation stimuli. Scale bar, 5 μm. **c** Quantification of total cAMP measured in trophozoites of dCas9-ctrl and AC2-g4159 cell lines using a cAMP ELISA assay at 1 h of encystation. **d** Quantification of *Gl*PKA-NBit luminescence in dCas9 control and AC2-g4159 cell lines at 0, 1, 2, and 4 h post exposure to encystation stimuli. The *Gl*PKA-NBit intensity is normalized to 0 h as percentage (%) of association. Decreased association indicates an increase of intracellular cAMP. **e** Relative protein level of MYB2-NLuc from dCas9-ctrl and AC2-g4159 cell lines at 0, 4, 8 h post exposure to encystation stimuli. **f** Western blot of CWP1 and tubulin from dCas9-ctrl and AC2-g4159 cell lines at 4 h post exposure to encystation medium. **g** Quantification of (**f**). All data are mean ± s.d. from three independent experiments. *P* values were calculated with two-tailed *t* tests.

characterized the morphology of these encysting cells and found 40% of the CWP1 positive encysting cells had developed ESVs up to Stage III[31] (Supplementary Fig. 9b, c), but none of these encysting cells differentiated into mature cysts. Together these results indicate that exogenous cAMP or AC2 overexpression is sufficient to drive the initiation of encystation in the absence of encystation stimuli, but does not support the completion of the differentiation program.

## Discussion
Encystation is a common developmental strategy employed for the dissemination and survival of stressful conditions by members of all eukaryotic supergroups[11]. While cAMP signaling is known to initiate

differentiation for survival in different environments[4–6], the molecular pathways underlying cAMP-induced stage transition across protozoan pathogens remain unclear. One challenge is the complexity of cAMP signaling in most alveolate and excavate parasites and another is that the lack of heterotrimeric G-proteins means that cAMP signaling is regulated differently than in model eukaryotes. *Giardia* has the smallest set of cAMP signaling components of any protozoan parasite making the study of cAMP signaling tractable in this organism. We have taken advantage of these features of *Giardia* and its well-established in vitro encystation to clarify the role of cAMP signaling.

The three common methods to activate encystation each result in reduced cholesterol availability[12]. A previously proposed hypothesis

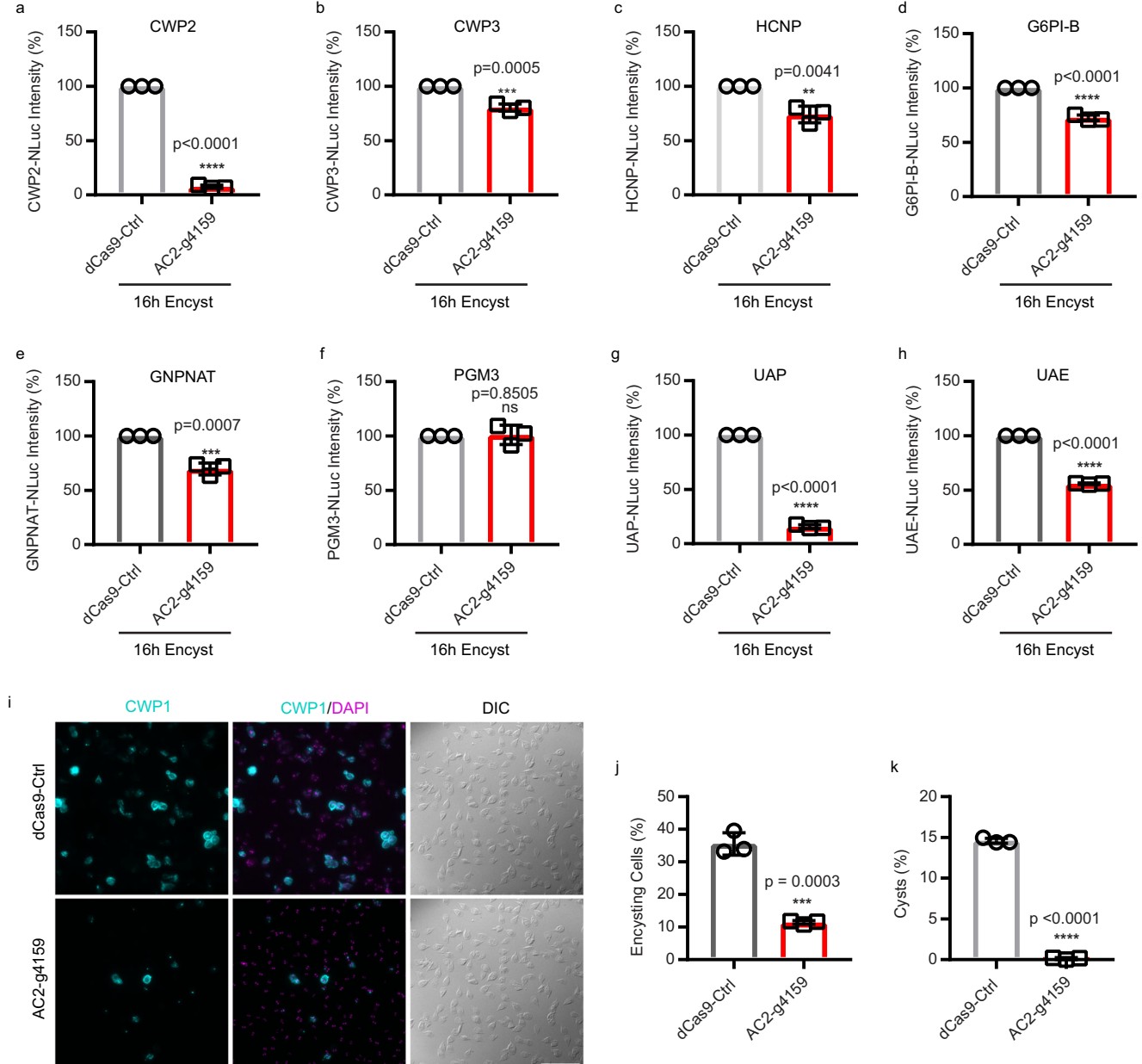

**Fig. 6 | AC2 knockdown impairs the upregulation of developmentally regulated cyst wall biosynthesis proteins. a–c** Relative expression levels of (**a**) CWP2-NLuc (GL50803_5535), (**b**) CWP3-NLuc (GL50803_2421), and (**c**) HCNCp-NLuc (GL50803_40376) in dCas9-Ctrl and AC2-g4159 knockdown cell lines. **d–h** Relative expression levels of GalNAc biosynthesis enzymes (**d**) G6PI-B-NLuc (GL50803_8245), (**e**) GNPNAT-NLuc (GL50803_14259), (**f**) PGM3-NLuc (GL50803_16069), (**g**) UAP-NLuc (GL50803_16217), and (**h**) UAE-NLuc (GL50803_7982) from dCas9-Ctrl and AC2-g4159 knockdown. **i, j** Representative

IFA images and quantification of 24 h encysting cells from dCas9 control and AC2-g4159 knockdown cell lines. Parasites were exposed to encystation medium for 24 h and stained with CWP1 antibody with DAPI (cells counted for dCas9-ctrl $n = 1265$, and AC2-g4159 $n = 1488$). Scale bar, 50 μm. **k** Quantification of mature cysts at 48 h post induction of encystation from dCas9 control and AC2-g4159 knockdown cell lines. Cyst counts were performed by hemocytometer (cysts counted for dCas9-Ctrl $n = 1432$, and AC2-g4159 $n = 12$. Data are mean ± s.d. from three independent experiments. $P$ values were calculated with two-tailed $t$-tests.

suggested that cholesterol binds a membrane cholesterol receptor and activates a sterol regulatory-element binding protein (SREBP) to modulate transcriptionally-regulated encystation[32]. However, candidates for the receptor or the SREBP have not been identified. We tested an alternative hypothesis in which membrane cholesterol depletion stimulates encystation by altering plasma membrane lipids. High cholesterol containing membrane microdomains, also known as lipid rafts, are generally understood to regulate signaling and previous work suggests they have a role in *Giardia* encystation[17,33,34].

Our results showed that treatments known from previous work to deplete plasma membrane cholesterol[19] elevated intracellular cAMP in

*Giardia* when applied to trophozoites and this in turn led to upregulation of encystation-specific genes. One goal of our study was to determine whether increased bile could directly alter plasma membrane cholesterol content, because bile-induced encystation occurs more rapidly than when parasites are induced to encyst with lipoprotein-deficient medium[12]. NR12S allowed us to monitor dynamic changes in membrane cholesterol because this probe indirectly reveals cholesterol levels[19]. Bile is a biological amphiphile and has been shown to directly interact with cholesterol and reorganize the lipid distribution of mammalian plasma membranes[18]. Several different bile salts have been shown to activate *Giardia* encystation with different

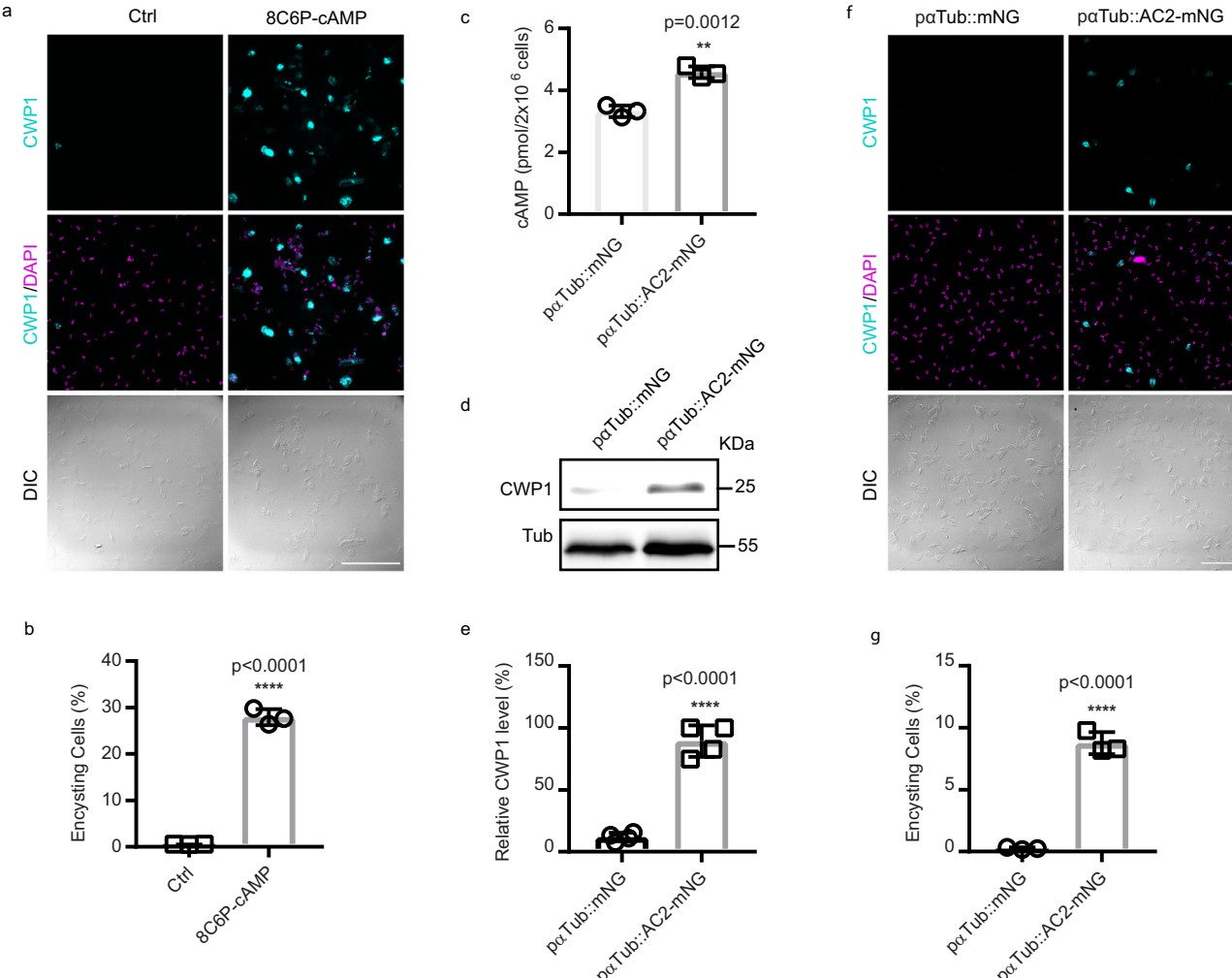

**Fig. 7 | Exogenous cAMP or AC2 over expression are sufficient to initiate encystation in the absence of encystation stimuli. a**, **b** Representative IFA images (**a**) and quantification (**b**) of encysting cells at 24 h post induction of 50 μM 8C6P-cAMP (total cells counted for Ctrl *n* = 1100, and 8C6P-cAMP *n* = 1221). Scale bar, 50 μm. **c** Quantification of intracellular cAMP from the indicated cell lines in growth medium. Western blot (**d**) and CWP1 quantification (**e**) of samples from parasites of the indicated cell lines. The blot is a representative of three independent experiments. Representative IFA images of encysting cells (**f**) and quantification (**g**) of indicated cell lines. Data are mean ± s.d. from three biological replicates using Student's *t* test (total cells counted for Ctrl *n* = 1187, AC2-mNG *n* = 1360, pαTub-mNG *n* = 1303, and pαTub-AC2-mNG *n* = 1221). Scale bar, 50 μm. Data are mean ± s.d. from at least three independent experiments. *P* values were calculated with two-tailed *t*-tests.

efficiencies[13]. MβCD, a water-soluble oligosaccharide with hydrophobic cavities, is widely used to deplete cholesterol from plasma membranes to study cellular signaling. However, MβCD is not specific for cholesterol[35], indicating that bile and MβCD could alter plasma membrane composition differently. On the other hand, our results show that alkaline pH enhances plasma membrane cholesterol extraction for both MβCD and bile.

Whether the increase of intracellular cAMP from the perturbation of plasma membrane fluidity is exerted indirectly due to changing plasma membrane lipid dynamics or by altering direct interactions between cholesterol and membrane receptors, or maybe both, has been a long-term debate[36,37]. It is now understood that the plasma membrane contains cholesterol and/or sphingolipid-enriched microdomains, where signaling can be activated or inhibited[38]. Plasma membrane cholesterol depletion selectively alters the amplitude of cAMP responses at lipid raft and non-raft domains indicating the compartmentalized nature of cAMP signaling at the plasma membrane[39]. Further elucidation of the relationship between lipid rafts and AC2 may provide insights into AC2 activation and the mechanisms used to sense bile salt alteration of the plasma membrane.

In this study, we generated a new cAMP signaling biosensor *Gl*PKA-NBit that reports on the activation of PKAc. NanoBit has been widely applied to study protein-protein interactions and generate genetic biosensors because of its reversibility and high luminescence. Comparison of cAMP kinetics from ELISA assays and the *Gl*PKA-Nbit reporter indicates that cAMP levels change more dynamically than are reported by *Gl*PKA-Nbit. This is likely because feedback mechanisms slow PKA dissociation. The NanoBit luminescent signal reported by the *Gl*PKA-NBit reflects the complex dissociation of PKA rather than acute changes in cAMP concentration, yet the biosensor is still valuable for revealing the initiation of cAMP signaling[40].

Our results provide clear evidence that encystation stimuli alter plasma membrane fluidity, activate AC2-dependent intracellular cAMP elevation accompanied with dissociation of PKAr and PKAc, and this increase of intracellular cAMP upregulates MYB2, CWPs, and four enzymes in the GalNAc pathway to enable successful cyst production. It is noteworthy that AC2 overexpression induces encystation but ESVs only progress to Stage III which is correlated with 6−8 h of encystation stimuli[31]. The lack of fully formed cysts reveals that additional signaling pathways that co-regulate differentiation remain to be discovered. The

identification of additional encystation signaling inputs and how plasma membrane cholesterol depletion activates AC2 awaits further investigation.

## Methods

### Plasmid construction

**mNeonGreen, Halo tag, and NanoLuc fusions.** Coding sequences were PCR-amplified from *Giardia lamblia* genomic DNA. Primers sequences are indicated in Supplementary Data 1. The mNeonGreen, Halo tag and NanoLuc parent vectors were digested with the indicated restriction enzymes and a PCR amplicon was ligated using Gibson assembly. The resulting constructs were linearized with the restriction enzyme indicated in Supplementary Data 1 before electroporation for integration into the endogenous locus[41]. Neomycin and puromycin were added for selection.

### Design of guide RNA for CRISPRi

Guide RNA design for the CRISPRi system utilized the Dawson Lab protocol[26], NGG PAM sequence and *G. lamblia* ATCC 50803 genome were selected for CRISPRi guide RNA design with Benchling.

### Design of *Gl*PKA-NanoBit

NanoBit PPI Starter System (N2014, Promega) was used to generate *Gl*PKA-NBit. Briefly, the complimentary peptide Small BiT (SmBiT; 11 amino peptide) was fused with PKAc (GL50803_101214) driven by its native promoter (~500 bp), and the Large BiT (LgBiT; 18 kDa) was fused to PKAr (GL50803_9117) driven by its native promoter (~500 bp) (Fig. 2c and Supplementary Fig. 2a). The control cell line expresses the PKAc-SmBiT and the LgBiT fragment driven by the PKAr native promoter (*p*PKAr) not fused to any protein (Supplementary Fig. 2a).

### *Giardia* growth and encystation media

*Giardia intestinalis* isolate WB clone C6 (American Type Culture Collection catalog number 50803) were grown in TYI-S33 media supplemented with 10% adult bovine serum and 0.125 mg/ml bovine bile at pH 7.1[42]. To induce encystation, cells were incubated for 48 h in pre-encystation media without bovine bile at pH 6.8 then incubated with TYI-S33 media at pH 7.8 supplemented with 10% adult bovine serum, 0.25 mg/ml porcine bile (B8631, Sigma-Aldrich) and 5 mM calcium lactate[13]. For studies of mature cysts cells were pre-encysted as above but then encysted using high bile method 2[43] modified to use 10 g/L ovine bovine bile (B8381, Sigma-Aldrich).

### cAMP analogs and inhibitor treatments

cAMP analogs and inhibitors, including Bromoadenosine 3′,5′-cyclic monophosphate sodium salt (8Br-cAMP, Sigma B7880), $N^6$,2′-O-Dibutyryladenosine 3′,5′-cyclic monophosphate sodium salt (DB-cAMP, TOCRIS Cat. No.: 1141), 8-(4-Chlorophenylthio)-$N^6$-phenyladenosine-3′,5′-cyclic monophosphate sodium salt (8C6P-cAMP, BIO-LOG Life Science Institute Cat. No.: C 043). Methyl-β-cyclodextrin (C4555, Sigma-Aldrich). 9-(Tetrahydron-2-furanyl)−9H-purin-6-amine, and 9-THF-Ade (SQ22536, C355, Sigma-Aldrich). N-[2-(p-Bromocinnamylamino) ethyl]−5-isoquinolinesulfonamine dihydrochloride (H89, B1427, Sigma-Aldrich), were used in this study. Cholesterol supplementation studies used 3β-Hydroxy-5-cholestene (C3045, Sigma-Aldrich).

For *Gl*PKA-NBit experiments (Fig. 2d, e), parasites were incubated 1 h with 50 μM 8C6P-cAMP, 10 μM H89, 10 μM SQ22536, 1 μM MβCD. For cAMP analogs and inhibitor treatments (Fig. 3a, Supplementary Fig. 3b, c, and Supplementary Fig. 8), parasite were incubated 1 h with 50 μM 8C6P-cAMP, 10 μM DB-cAMP, 10 μM 8Br-cAMP, and 10 μM SQ22536 in pre-encystation medium at pH 6.8, washed out, and encysted 4 h and 24 h with encystation medium at pH 7.8.

### Live imaging and plate reader assay of NR12S and CTXB

3-((3-((9-(Diethylamino)−5-oxo-5*H*-benzo[*a*]phenoxazin-2-yl)oxy)propyl)(dodecyl)(methyl)ammonio)propane-1-sulfonate (NR12S, Cat No 7509, Tocris) was used to measure membrane fluidity. Cholera toxin subunit B, Alexa Fluor™ 594 conjugate (CTXB, Invitrogen, C34777) was used to stain lipid rafts. For live imaging, parasites were incubated 60 min with or without treatments in a Panasonic tri-gas incubator set to 5% Co2 and 2% O2. Before imaging, parasites were incubated 30 min with 0.05 μM NR12S or 5 μg/mL CTXB after exposure to different treatments, washed out with 1X HBS (HEPES-buffered saline). Images were acquired on a DeltaVision Elite microscope using a 60X, 1.42-numerical aperture objective with a PCO Edge sCMOS camera, YFP and mCherry filter set, and images were deconvolved using SoftWorx (API, Issaquah, WA).

For plate reader assay, cells were resuspended in cold 1X HBS and dilutions were made after measuring cell density using a MOXI Z mini Automated Cell Counter Kit (Orflo, Kenchum, ID). To measure NR12S fluorescence intensity, 20,000 cells were loaded into black polystyrene, clear bottom 96-well plates (Greiner). The fluorescence intensity of NR12S were measured using a TECAN SPARK plate reader with excitation wavelength 520 nm, and emission wavelength 560/630 nm.

### ELISA assays

Encysted *Giardia* cells were iced for 15 min and centrifuged at 700×*g* for 7 min at 4 °C. Cells were resuspended in cold 1X HBS and dilutions were made after measuring cell density with a MOXI Z mini Automated Cell Counter Kit (Orflo, Kenchum, ID). To measure intracellular cAMP levels, $2 \times 10^6$ cells were mixed with 0.1 M HCl and incubated at room temperature for 20 min. The supernatant was mixed with ELISA buffer and loaded into 96-well plates provided by Cayman chemical (Cyclic AMP ELISA Kit, 581001, Cayman Chemicals). Briefly, the cAMP standard was generated with ELISA buffer. Both sample and standard were acetylated individually to ensure maximum sensitivity. cAMP ELISA antiserum and tracer were added into 96 wells plate and incubated 18 h at 4 °C. The next day, 200 μl Ellman's reagent was added and incubated 2 h. The absorbance 405–420 nm was read using TECAN SPARK plate reader. These absorbance readings were used to determine the sample concentration according to the manufactures instructions.

### In vitro bioluminescence assays

*Giardia* cells were iced for 15 min and centrifuged at 700×*g* for 7 min at 4 °C. Cells were resuspended in cold 1X HBS and dilutions were made after measuring cell density with a MOXI Z mini Automated Cell Counter Kit. To measure NanoLuc luminescence, 20,000 cells were loaded into white polystyrene, flat bottom 96-well plates (Corning Incorporated, Kennebunk, ME) then mixed with 10 μl of NanoGlo luciferase assay reagent (Promega N2012). Relative luminescence units (RLU) were detected on a pre-warmed 37 °C EnVision plate reader (Perkin Elmer, Waltham, MA) for 30 min to reach the maximum value. Experiments are from three independent bioreplicates.

### Protein blotting

*Giardia* parasites were iced for 30 min then centrifuged at 700×*g* for 7 min and washed twice in 1X HBS supplemented with HALT protease inhibitor (Pierce) and phenylmethylsulfonyl fluoride (PMSF). The cells were resuspended in 300 μl of lysis buffer contains 50 mM Tris-HCl pH 7.5, 150 mM NaCl, 7.5% glycerol, 0.25 mM CaCl$_2$, 0.25 mM ATP, 0.5 mM Dithiothreitol, 0.5 mM PMSF, 0.1% Triton X-100 and Halt protease inhibitors (Pierce). The sample was pelleted at 700×*g* for 7 min, the supernatant was mixed with 2X sample buffer (Bio-Rad) and heated at 98 °C for 5 min. Protein samples were separated using sodium dodecyl sulfate (SDS) polyacrylamide gel electrophoresis. Protein was transferred to Immobilon-FL membrane (Millipore). To detect tubulin, a mouse monoclonal anti-acetylated tubulin clone 6-11B-1 antibody

(IgG2b; product T 6793; Sigma-Aldrich) were used at 1:2500 and secondary anti-IgG2b mouse isotype-specific antibody conjugated with Alexa 488 (Molecular Probes A-21141) were used at 1:2,500. To detect CWP1, Alexa 647-conjugated anti-CWP1 antibody (Waterborne, New Orleans, LA, A300AF647-R-20X) was used at 1:2,000. Multiplex immunoblots were imaged using a Chemidoc MP system (Bio-Rad). Full-length blots are provided in Supplementary Data 2.

## Immunofluorescence

*Giardia* parasites were iced for 30 min and pelleted at 700 x g for 7 min. The pellet was fixed in PME buffer (100 mM Piperazine-*N,N'*-bis (ethanesulfonic acid) (PIPES) pH 7.0, 5 mM EGTA, 10 mM MgSO₄ supplemented with 1% paraformaldehyde (PFA) (Electron Microscopy Sciences, Hatfield, PA), 100 μM 3-maleimidobenzoic acid *N*-hydroxysuccinimide ester (Sigma-Aldrich), 100 μM ethylene glycol bis (succinimidyl succinate) (Pierce), and 0.025% Triton X-100 for 30 min at 37 °C. Fixed cells were attached on polylysine-coated coverslips. Cells were washed once in PME and permeabilized with 0.1% Triton X-100 in PME for 10 min. After two quick washes with PME, blocking was performed in PME supplemented with 1% bovine serum albumin, 0.1% NaN₃, 100 mM lysine, and 0.5% cold water fish skin gelatin (Sigma-Aldrich). Next, 1:200 diluted A300-Alexa 647-conjugated anti-CWP1 antibody was incubated for 1 h at 1:200. Cells were washed three times in PME plus 0.05% Triton X-100. Coverslips were mounted with Pro-Long Gold antifade plus 4′,6-diamidino-2-phenylinodole (DAPI; Molecular Probes). Images were acquired on a DeltaVision Elite microscope using a 60X, 1.42-numerical aperture objective with a PCO Edge sCMOS camera, and images were deconvolved using SoftWorx 7.0.0 (GE Healthcare, Issaquah, WA).

## Image analysis

Fiji/ImageJ 2.15.0 was used to process all images. Figures were assembled using Adobe Illustrator CS6.

## Cyst count and cyst viability staining

*Giardia* trophozoites were cultured 24 h in high bile encystation media (above) then continued with TYI-S33 for another 24 h. After 48 h the total cell number was determined with a MoxiZ coulter counter (Orflo). The encysted culture was centrifuged at 700×*g* for 7 min and the pellets were washed 10 times in deionized water, then stored in distilled water overnight at 4 °C. To count cyst concentration, 20 μl of 48 h encysted cells were counted using a hemocytometer. To determine cyst viability, fluorescein diacetate (FDA) and propidium iodide (PI) were used to stain live and dead cysts and images were collected using a DeltaVision Elite microscope with a 40X, 1.2-numerical aperture objective with a PCO Edge sCMOS camera, and images were deconvolved using SoftWorx 7.0.0 (GE Healthcare, Issaquah, WA).

## Statistical analysis

All statistical analyses were performed using Prism software version 9.4.1 (GraphPad Software Inc., CA, USA). All reported p-values are from pairwise analysis using a two-tailed t-test. *P*-values < 0.05 were considered significant. Significant differences are indicated as follows: *$P \le 0.05$, **$P \le 0.01$, ***$P \le 0.001$, ****$P \le 0.0001$. All microscopy images are representatives of at least three independent experiments and all experiments resulted in comparable results.

## Reporting summary

Further information on research design is available in the Nature Portfolio Reporting Summary linked to this article.

## Data availability

Data supporting the findings of this work are available within the article and its Supplementary Information files. Raw values used for graphs and uncropped blots are in Source Data. All *Giardia* DNA sequences used in this study are available at giardiadb.org. Source data are provided with this paper.

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

## Acknowledgements

We thank B. Wakimoto, J. Kollman, and members of the Paredez lab for discussion and edits. Research was funded by NIH R21AI159035 and R01AI168417 awarded to A.R.P.

## Author contributions

A.R.P. and H.W.S. conceived and designed the study. H.W.S. and G.C.M.A. performed experiments and analyzed data. A.R.P., H.W.S. and G.C.M.A. produced the figures. A.R.P. and H.W.S. wrote the manuscript. A.R.P. secured the funds. All authors reviewed and approved the final version of the manuscript.

## Competing interests

The authors declare no competing interests.
