## [Peer Review File · Nature Communications]

Reviewers' Comments:

Reviewer #1:

Remarks to the Author:

Han-Wei Shih et al have studied the role of cAMP in the encystation of Giardia. It is not the first time cAMP has been shown to be up-regulated during early stages of Giardia cyst formation and it not the first time the two adenylate cyclases in Giardia have been described and the down- stream effector PKA has earlier been shown to affect encystation. However, this is the first time anyone studies these things at the same time with different approaches and a large amount of data is generated. I clearly shows a role of cAMP in induction of Giardia encystation and the main players are studied. This is important not only for the Giardia research field but also for other cyst-forming protozoa, that are abundant as parasites and free-living organism. I have the following specific comments.

Line 24. I would mention cholesterol starvation here already and reference Lujan.

Line 33. PKA is also important for excystation of Giardia, so the complete differentiation process is regulated by PKA, see doi: 10.1074/jbc.M208033200.

This paper also shows an effect of calcium levels, something that affects adenylate cyclases and encystation efficiency.

Line 35. Also many free-living like Dictyostelium discoideum.

Line 47. It would be interesting to see phylogenetic trees of these enzymes in Giardia (and other Diplomonads) involved in cAMP metabolism to see how they have evolved.

Line 50. How is it in other intestinal protozoa, that forms resistant forms like Entamoeba and Cryptosporidium, what do the complement of protein look like?

Line 59. In original studies 5mM lactate stimulated encystation. Was that included here? It is possible that the original effect seen by lactate actually was due to that it was a Ca²⁺ salt and that 5mM Ca²⁺ stimulate encystation rather than lactate. Calcium signalling is often associated with cAMP levels, see eg ref 4. What were the calcium levels here?

Line 71. Methyl-B-cyclodextrin removes lipid rafts and they have been shown by Sid Das to be important in encystation. This can be mentioned here already.

Line 73. Were conditions mimicking the high bile protocol tested here?

Line 86. The panels with representative figures in Fig. 1D do not fit with the bars in Fig. 1E. There are more red cells in M β CD pH 7.8 than in the bile panel but the bar in Fig. 1E is lower for bile than for M β CD. I also think that there is a quite big diversity in these two panels compared to the control. The control has green cells but the mid and right panel has green, yellow and red cells. This is not reflected in the data in the bars, can it be presented in another way.

Line 92. The cholesterol in Giardia is found in lipid-rafts, see

DOI: 10.3389/fcimb.2022.974200

doi: 10.1128/IAI.03118-14. Rafts are important in encystation and less cholesterol can give less rafts and encystation is disturbed.

GM1- cholera toxin B binding, is used to label lipid rafts in Giardia. Can some type of co-labelling be used here? Also alter in the AC2 localization.

Line 96, can mention again that PKA is important for encystation here.

Line 104-108 can be deleted.

Line 110. In Gibson et al Western blots were used with anti-gPKAr and -gPKAc rabbit polyclonal antibodies. The experiment was done in parallel and PKAc was very constant through-out the experiment, whereas PKAr was down-regulated, generating a stable degradation product.

They also used the high bile method of encystation described by Kane et al, which is different than what was used here. The problem with luciferase measurements in cells, as used here, is that you do not know if you are measuring a full length PKA protein or a fragment containing luciferase. In Gibson et al a clear stable degradation product is form from PKAr. Ask for the PKA antibodies and use that in Westerns. You can also run Western blots instead of luciferase measurements.

Line 146. What encystation stimuli.

Line 154, this result is in line with earlier results, can be cited.

Line 157. The figure legends of Sup. Fig 3 and 4 were switched.

Line 160. How was the washing done? If the high bile protocol is used there is no need for pre-encystation incubation but with the protocol used here the efficiency is much reduced if no true pre-encystation is done.

Line 167. Sup. Fig. 3E show a large number of dead cysts, is this a DMSO effect? How much DMSO was used? Make an encystation without DMSO and use that as control also. Later data show that increased AC2 activity and cAMP production do not produce more mature cysts, how can you explain this difference?

Line 171. Write the WB ORF numbers here also to clarify. The ORF numbers of all cAMP related proteins can be added to Sup. Fig. 1.

Line 176. Again, something is wrong with the figure legends in Sup Fig 3 and 4. Was AC1 localized in 12 and 20h to see the dynamics?

Can the small vesicles seen by Benchimol with GalNAc or small vesicles in Einarsson et al 2016 be the same as the ones here?

Line 181. Co-localize choleraToxin B with AC2 to check lipidraft localization. AC2 shows up in Sid Das proteomics data set from lipid rafts.

Line 184. There is quite some expression data on both RNA and protein level during different conditions in Giardia DB now. Does that say something about the functions about the cAMP related proteins?

Line 194. What happens with AC1 expression when AC2 knocked down? What happens with the levels of the PDE when AC2 is knocked down? Were PDE and AC1 also knocked down?

Line 226. Counting ESVs/cell at different time points of encystation is a better measurement of how encystation proceeds compared to CWP-1 expression. Was that done?

Line 230. Can these more viable cysts excyst? Can the cAMP rescued AC knockdown cysts excyst? This would be a good measurement of that real cysts were produced in the process.

Line 240. Again, very low level of viable cysts?

Line 245. It looks like exogenous cAMP can do it in a few cells but not all, depending on cell cycle stage?

Line 332: Ref lacking. Can the sequence of the functional guide RNA be added here or elsewhere?

Line 433. Why was this protocol used? Ovine bovine bile does not work as well as only bovine. Calcium lactate is usually not used in the high bile protocol.

Reviewer #2:

Remarks to the Author:

The current manuscript entitled the "Encystation stimuli sensing by adenylate cyclase AC2-dependent cAMP signaling in Giardia" is interesting and highly innovative. It describes the molecular mechanism of giardial encystation via adenylate cyclase/cAMP pathways.

Giardiasis, caused by an intestinal protozoan, *Giardia lamblia*, is a major public health problem worldwide. Encystation or the formation of water-resistant cysts allows *Giardia* to survive in the environment for months before transmitting to a new host. Although the development of methodologies for in vitro encystation has revolutionized the research on *Giardia* over the past 40 years, the actual mechanism of encystation is still elusive. In the current manuscript, using state-of-the-art molecular technologies, Shih et al. have nicely demonstrated that adenylate cyclase (AC) plays a critical role in sensing encystation stimuli generated by high bile and high pH conditions. The authors also demonstrated that high-bile and higher pH (7.8) stimulates cAMP levels by activating AC2 enzymes that causes the re-localization of cAMP-activated protein kinase enzymes.

In Fig.1, the authors demonstrated that bile treatment depletes cholesterol from the plasma membranes of *Giardia* and alters the membrane fluidity that could be responsible for inducing encystation. In this context, the bile is effective than the higher pH (7.8) alone, however, the combination of bile and high pH is more efficient. The membrane fluidity was assessed using NRI 2S

membrane binding dye and encystation was monitored by assessing the induction of CWP1. To support the results bile-induced cholesterol depletion, the authors treated cells with MBCD, which is a cholesterol-binding agent and lowers cholesterol from membranes.

My question is: did the authors see any changes of membrane structures after bile and MBCD treatment? Did the investigators try to neutralize the effects of bile and MBCD by adding excess cholesterol from outside?

In Fig.3, the use of membrane permeable cAMP (86P-cAMP) was used to assess the induction of encystation. Although the data is convincing, images shown in panel c are somewhat blurry, especially the DIC images. Authors might consider replacing the current image with improved one. Also, is it possible to show the images in higher magnification? Did authors try to assess the ESV biogenesis side by side with CWP1 expression? This should confirm the link between CWP expression and encystation as ESV biogenesis is the hallmark of cyst formation (encystation).

Fig.4. It appears that AC2 is involved in encystation and cyst production. However, the rationale for not participation of AC1 in encysting process requires additional discussion. Authors might consider showing AC1 localization and encystation (SP Fig.3) side by side with AC2 in Fig.4 (main text).

Fig.5. The results on AC2 knockdown experiment impairs the encystation process is interesting. However, it will be more convincing if the authors show the results of AC1 knockdown and compare that with AC2 knockdown.

Fig.6. shows that the addition of cAMP or the increased level AC2 is sufficient for initiation of the encystation process. Is it possible then a change in the microenvironment of Giardia causing excess cAMP production should also trigger encystation? What would happen if AC1 produces excess cAMP? In addition, do the authors predict the presence of an encystation-related receptor in Giardia that is likely to modulate AC2 activation?

SP Fig.1 is informative. Authors might consider incorporating this figure of predicting cAMP signaling pathway in Giardia in the main text.

Overall, this manuscript provides an excellent and innovative approach to study the role of high bile induced plasma membrane alteration, adenylate cyclase activation/cAMP production in inducing encystation by Giardia.

Reviewer #3:

Remarks to the Author:

This work postulates that cAMP synthesis mediated by the AC2 enzyme would be necessary to initiate the encystment process in the intestinal parasite Giardia lamblia.

In this work, the authors postulate that the induction of encystment of G. lamblia (through the use of Bile or MBCD), produces cholesterol depletion of the plasma membrane, affecting the composition of the lipids-raft domains and its fluidity. The latter could trigger a signal transduction response, as has already been described in other cell models.

In this context, and based on the observation that cAMP levels in G. lamblia increase during the first hour of encystment, the authors speculate that this signal could be mediated by one or both of the adenyl cyclases of G. lamblia.

Next, they found that AC2 (an endogenously tagged form of AC2) is located mainly in the plasma membrane, while AC1 was found in structures similar to intracellular vesicles. They also describe that the expression of AC1 increases 8 to 16 hours after the encystment stimulus, while the expression of

AC2 remains constant throughout the process.

From this moment, the authors focus on AC2 as responsible for the cAMP increased after encystment induction.

In summary, the authors conclude that cholesterol depletion over the plasma membrane of trophozoites affects its fluidity and increase cAMP synthesis by AC2 located in plasma membrane. This increase in cAMP produces the uncoupling of the PKAr-PKAc complex and the subsequent expression of proteins linked to encystment, such as MYB2 and CWPs among others, and the consequent generation of viable cysts.

Finally, through over expression of AC2 and/or the use of non-hydrolyzable cAMP analogues, the authors observe that -without mediating encystment stimulus- there is an increase in trophozoites that initiate the encystment process, but do not generate cysts, indicating that the cAMP signal transduction pathway is necessary to initiate the encystment process but not sufficient to fulfill it.

Methods used

Trophozoite culture, generation of stable lines expressing endogenously marked proteins, incubation with permeable and non-hydrolyzable cAMP analogues, generation of AC2 knockdown using the CRISPRi technique, immunofluorescence microscopy, among others.

It is important to highlight the development and implementation, in *G. lamblia*, of the novel PKA activation sensor using the NanoBit system.

In general terms, the work seems very interesting to me, the experiments are well planned, and their results are complementary to each other.

However, I have some observations and doubts that I would like the authors to take into account:

Inducers of membrane cholesterol depletion trigger cAMP signaling

>In rows 125 to 127 the authors state that the Small BiT (SmBiT) complementary peptide was fused with PKAc driven by its native promoter, and the Large BiT (LgBiT) was fused to PKAr driven by its native promoter. I understand that in this case, the proteins were not "endogenously tagged"? As in the case of PKAc-NanoLuc and PKAr-NanoLuc (lines 112) or with mNeonGreen and the Haloalkane dehalogenase (lines 117).

What is this difference due to? It is a transient expression?

>Figure 2d and e: DMSO control have no error bar. Do I have to interpret that it is very small and cannot be seen? The same does not happen in figure SP Figure 2b, where it can be seen that the result of the measurement of trophozoites that express GLPKA-NBit (Equivalent to the control of figure 2d and e) presents a dispersion of $\pm 8.3\%$.

Even when the control is set to 100%, this 100% should represent the average of the samples and take into account the dispersion of the measurements. How many samples does each condition have?

>The same observation can be made for all the figures with similar experiments.

>Lines 149 to 151: At this point, I would say that the increase in membrane fluidity by treatment with M β CD or Bile triggers the elevation of intracellular cAMP and the activation of PKAc. I don't know if I would ascribe the entire effect to cholesterol depletion. The effect of treatment with Bile is more intense than that of M β CD and bile should be less specific in lipid uptake. M β CD isn't 100% specific to cholesterol, neither.

Exogenously supplied cAMP upregulates encystation rates

>The legends of the figures SP 3 and SP 4 are interchanged with each other.

AC2 is necessary for cyst production

>Line 211: The parentheses are not closed correctly

>Line 212 to 214: At what time after encystment was the expression of CWP2, CWP3, HCNCP, and the five GalNAc measured? I cannot find this data in the text or in the legend of figure 5.

AC2 is necessary for the initiation of cAMP-dependent encystation

>In figure 2d and 2e, the activation of PKA_Nbit is plotted as a decrease in luminescence, while in figure 4c a similar experiment is shown differently (% of dissociation). I believe that the way in which the results of similar experiments are presented should be consistent in order to avoid confusion.

>The western blot of Supplementary Figure 3b is not representative of what is seen in Supplementary Figure 3c. I analyzed the wb with the program imagej, and the behavior presented in the bar graph does not match what is seen in the Western blot.
How many images were analyzed for this result?

>In Supplementary Figure 3a (sp fig. 3a), the amount of total cAMP is shown in pmol/ml. Is this the concentration of cAMP within the trophozoite or is this the concentration of the ELISA reaction mix? In materials and methods, it is specified that 20,000 cells per test were used, but in this figure it is marked as 2×10^6 cells (please check).
I think the amount of cAMP should be expressed in pmol/20000 cells or similarly.

> Lines 189 to 190: Are the same experiments of sp fig. 3a done with the clone AC2-g4159? Figure 4b shows only one point in time. Are these AC2-g4159 trophozoites in figure 4b incubated in normal medium or in encystation medium?

Materials and Methods

mNeonGreen, Halo tag and NanoLuc fusions

>It took me a bit to understand the procedure for the construction of the Plasmids and the strategy in general. Would a slightly more detailed description be possible with some diagram similar to what is shown in figure sp1? One of the most interesting aspects of this work was this, and the details of its realization are scarce.

>Line 329: I can't find the restriction enzymes supposedly listed in Supplemental File 1.

>Line 330 reference 38: I think this reference is more appropriate to explain the strategy used.
<https://journals.asm.org/doi/10.1128/EC.00190-10>

Design of GLPKA-NanoBit

>Line 335 to 341: >Are these transient transfections?

Discussion

After demonstrating that AC1 is not located in the plasma membrane and that it is preferentially expressed between 8 and 16 hours after the onset of encystment, AC1 is ignored by the authors. It is interesting to note that in the authors' words, *G. lamblia* is an organism with a cAMP machinery with minimal redundancies (Line 43), and while AC1 may not be playing a role in the encystment process,

it is hard for me to ignore the fact that AC1 expression is being induced right during this process, as are MYB2, CWPs, and GalNAcs.

Is it possible that AC1 expression is regulated by AC2? What would happen with the expression of AC1 in the clone AC2-g4159? Is it possible that AC1 expression is a necessary and independent step of AC2 to produce viable cysts?

We thank the reviewers for their careful evaluation of our manuscript. Addressing their comments has helped refine the manuscript and make the work more approachable for a broad audience. All major points have been addressed.

Editor	Authors Response
In particular, we ask that you address Referee 2's comments relating to the need to confirm the link between CWP expression and encystation, and to show AC1 and AC2 localisation. We also ask that you address the concerns raised by Referee 3 relating to the text, conclusions and figure editing.	
Reviewer #1	
Line 24. I would mention cholesterol starvation here already and reference Lujan.	Done.
Line 33. PKA is also important for excystation of Giardia, so the complete differentiation process is regulated by PKA, see doi: 10.1074/jbc.M208033200. This paper also shows an effect of calcium levels, something that affects adenylate cyclases and encystation efficiency.	We have added that PKA is also important for excystation. We appreciate that reviewer 1 brought up the possible involvement of Ca²⁺ signaling in cAMP signaling, which is a topic we would like to explore in the future.
Line 35. Also many free-living like Dictyostelium discoideum.	We now mention Dictyostelium discoideum where the role of cAMP has been well studied. .
Line 47. It would be interesting to see phylogenetic trees of these enzymes in Giardia (and other Diplomonads) involved in cAMP metabolism to see how they have evolved.	We agree that this is potentially interesting, however we can't imagine any result that would alter the interpretation of our study and the evolution of these enzymes is beyond the scope of our manuscript.
Line 50. How is it in other intestinal protozoa, that forms resistant forms like Entamoeba and Cryptosporidium, what do the complement of protein look like?	This is an interesting question. We considered performing extensive analysis for a table before the reviewer asked about this but then decided that would be more appropriate for a review. Nevertheless, our statement that Giardia has a simple cAMP signaling system that is less complex than in other protists holds true: E histolytica is far more complex (4 GPCRs, 6 PDEs, 8PKAs, etc. See PMID: 33102254. C. parvum is slightly more complex with 3 PKAs, PMID: 21962082 and 3 adenylate cyclases (BLAST search of CryptoDB).
Line 59. In original studies 5mM lactate stimulated encystation. Was that	Yes, 5mM Calcium lactate was included. A quick and dirty experiment suggests that both Ca²⁺ alone and lactate

included here? It is possible that the original effect seen by lactate actually was due to that it was a Ca ²⁺ salt and that 5mM Ca ²⁺ stimulate encystation rather than lactate. Calcium signalling is often associated with cAMP levels, see eg ref 4. What were the calcium levels here?	alone help promote encystation, but this needs more careful testing. We plan to examine the involvement of Ca ²⁺ in the future. To fully address the issue will require many experiments that are beyond the scope of this paper.
Line 71. Methyl-B-cyclodextrin removes lipid rafts and they have been shown by Sid Das to be important in encystation. This can be mentioned here already.	We have cited the Das lab article PMID: 25733521
Line 73. Were conditions mimicking the high bile protocol tested here?	Yes. We analyzed CTXB and NR12S after high bile treatment and observed changes in membrane fluidity as you would expect.
Line 86. The panels with representative figures in Fig. 1D do not fit with the bars in Fig. 1E. There are more red cells in MβCD pH 7.8 than in the bile panel but the bar in Fig. 1E is lower for bile than for MβCD. I also think that there is a quite big diversity in these two panels compared to the control. The control has green cells but the mid and right panel has green, yellow and red cells. This is not reflected in the data in the bars, can it be presented in another way.	First we want to point out that the NR12S values in the graph are from plate reader assays with 20,000 cells/well. So, the values are averages of the entire population. The images that we presented are fields of view from our fluorescent microscope, the images are qualitative as no measurements were made from these. Since the number of cells in an individual field of view is much smaller than what we measured in the plate reader it is possible that a single field of view could diverge from the mean. We replaced the original image with another that perhaps better aligns with our quantification of the MβCD treatment.
Line 92. The cholesterol in Giardia is found in lipid-rafts, see DOI: 10.3389/fcimb.2022.974200 doi: 10.1128/IAI.03118-14. Rafts are important in encystation and less cholesterol can give less rafts and encystation is disturbed.	We cite PMID: 25733521 and the new Supplemental Figure 1 specifically demonstrates this.
GM1- cholera toxin B binding, is used to label lipid rafts in Giardia. Can some type of co-labelling be used here? Also alter in the AC2 localization.	As reviewer suggested, we colocalized AC2-mNG with CTXB new Supplemental Figure 1. We found that AC2-mNG colocalized with CTXB (lipid rafts). We observed lipid raft were depleted in MβCD and encystation treatments, but we did not see any obvious change in AC2-mNG localization.
Line 96, can mention again that PKA is important for encystation here.	We have mentioned the importance of PKA as suggested.
Line 104-108 can be deleted.	We kept line 104-108 to ensure a broad audience can understand the principle behind NanoBit.
Line 110. In Gibson et al Western blots were used with anti-gPKAr and -gPKAc rabbit polyclonal antibodies. The	Point taken, we agree that it is technically possible we only quantified the presence of NanoLuc and a western blot is necessary to exclude this possibility. We previously

experiment was done in parallel and PKAc was very constant through-out the experiment, whereas PKAr was down-regulated, generating a stable degradation product. They also used the high bile method of encystation described by Kane et al, which is different than what was used here. The problem with luciferase measurements in cells, as used here, is that you do not know if you are measuring a full length PKA protein or a fragment containing luciferase. In Gibson et al a clear stable degradation product is form from PKAr. Ask for the PKA antibodies and use that in Westerns. You can also run Western blots instead of luciferase measurements.	requested the antibody from Dr. Chakrabarti but never got a response. Fortunately, our integrated NanoLuc reporter is fused with 3HA allowing us to follow protein levels by western blotting. Gibson (PMID: 16472811) showed the PKAr is degraded at 2 and 6 h exposure to encystation stimuli compared to 0h, but western blots performed in response to this question indicates that there is no change at 2 and 6 hours (SP Fig 2a-b). As stated before Gibson did not include a loading control and Giardia expresses a large number of proteases that can make preventing non-specific protein degradation challenging.
Line 146. What encystation stimuli.	We now specify that high bile encystation medium was used. Additionally, we more clearly defined the encystation protocols used in the methods.
Line 154, this result is in line with earlier results, can be cited.	We explicitly stated that PKA relocates in response to encystation stimuli (PMID: 16472811). Citing this again on line 154 when discussing G/PKA-NBit might make the readers believe our reporter is not novel.
Line 157. The figure legends of Sup. Fig 3 and 4 were switched.	We apologize for the mistake, it has been corrected.
Line 160. How was the washing done? If the high bile protocol is used there is no need for pre-encystation incubation but with the protocol used here the efficiency is much reduced if no true pre-encystation is done.	We used the two-step encystation protocol for cAMP analog treatment experiments. So the washing was done in pre-encystation medium before exposing to two step encystation medium (porcine bile). However, we also indicate in the methods section that we generally pre-encyst cells because we think it helps synchronize the process.
Line 167. Sup. Fig. 3E show a large number of dead cysts, is this a DMSO effect? How much DMSO was used? Make an encystation without DMSO and use that as control also. Later data show that increased AC2 activity and cAMP production do not produce more mature cysts, how can you explain this difference?	We used 0.005% DMSO which we know is below the threshold for inhibiting Giardia proliferation PMID: 2867963. As the cAMP analogs must be diluted in DMSO and a control should only have one experimental condition altered we have already included the proper control. Figure 3e (Sup Fig 3 is AC1 expression) showed that a 1hr pulse of cAMP before adding encystation stimuli upregulated cyst formation. The AC2 over expression studies you are referring to were performed in growth medium (not encystation medium) to test if AC2 alone could trigger encystation. We did not test AC2 over

	expression in encystation medium because we cannot over express AC2 for only 1 hour to directly compare with a pulse of exogenous cAMP.
Line 171. Write the WB ORF numbers here also to clarify. The ORF numbers of all cAMP related proteins can be added to Sup. Fig. 1.	Done, now Fig 1 in current version.
Line 176. Again, something is wrong with the figure legends in Sup Fig 3 and 4. Was AC1 localized in 12 and 20h to see the dynamics?	The legends are now corrected. We include images for 0,4, 8, 16, and 24h.
Can the small vesicles seen by Benchimol with GalNAc or small vesicles in Einarsson et al 2016 be the same as the ones here?	We don't know but would be happy to share our AC1-mNG construct or cell line with any lab interested in following up on this.
Line 181. Co-localize choleraToxin B with AC2 to check lipidraft localization. AC2 shows up in Sid Das proteomics data set from lipid rafts.	We now show that AC2-mNG partially colocalized with CTXB which is consistent with the Das lab proteomic dataset (SP Fig 1c).
Line 184. There is quite some expression data on both RNA and protein level during different conditions in Giardia DB now. Does that say something about the functions about the cAMP related proteins?	RNAseq studies are excellent for hypothesis generation, but observations need to be validated with wet lab experiments examining protein levels. Some of what you ask is addressed in the next response.
Line 194. What happens with AC1 expression when AC2 knocked down? What happens with the levels of the PDE when AC2 is knocked down? Were PDE and AC1 also knocked down?	As reviewer suggested, we transformed AC2 g4159 into AC1-NLuc and PDE-NLuc. Our results showed that PDE expression level is reduced but the level of AC1 is not impacted. AC1 is upregulated many hours after AC2 so they appear to be independent of each other.
Line 226. Counting ESVs/cell at different time points of encystation is a better measurement of how encystation proceeds compared to CWP-1 expression. Was that done?	Although ESV counts are informative and used by us in previous studies, ESV number cannot represent the whole encystation process because they only represent CWP1-3. Instead, we decided to examine the expression level of all CWPs and GalNAc enzymes which we considered a more inclusive approach.
Line 230. Can these more viable cysts excyst? Can the cAMP rescued AC knockdown cysts excyst? This would be a good measurement of that real cysts were produced in the process.	Answering this question would require a lot of work for what we believe is a minor point. The thrust of this work was to examine how encystation is induced. Further we do show that cAMP alone is not sufficient for inducing fully formed cysts so more awaits to be discovered.
Line 240. Again, very low level of viable cysts?	We don't know how to address this comment besides saying that encystation rates vary with strains and even within a single lab. This is discussed in Chapter 23 of the Giardia A model organism book. What is important is that we directly compared experimental manipulations of the same parental cell line and found differences.

Line 245. It looks like exogenous cAMP can do it in a few cells but not all, depending on cell cycle stage?	In preparing this revision we re-organized our raw data for submission to figshare and discovered that there was a mixup in the reported results. We reported the results for PDE knockdown rather than exogenous cAMP. Knockdown of PDE results in 7% of cells entering the encystation program a result we plan to include in a future study. A pulse of exogenous cAMP activated encystation in 27.9% of cells, the text and figures have been corrected. This exceeds what we see for AC2 over expression and likely reflects the need for a transient pulse of cAMP that is not possible through over-expression of AC2 using the currently available genetic tools for Giardia. We did not examine the cell cycle of CWP1 positive cells, but AC2 levels are higher in G2/M according to a study from Janet Yee's group.
Line 332: Ref lacking. Can the sequence of the functional guide RNA be added here or elsewhere?	The reference for CRISPRi has been added. Sequences for all of the primers including gRNA sequences used in this study are in the supplemental excel file.
Line 433. Why was this protocol used? Ovine bovine bile does not work as well as only bovine. Calcium lactate is usually not used in the high bile protocol.	The protocols used in this study are from Giardia A model organism book. Method 1 is the two-step method and Method 2 is a high bile method originally published by Sun 2003. Although Sun 2003 indicates bovine bile, the Giardia book version indicated ovine bovine bile. We reduced the 12.5 g/l to 10g/l because it works similarly and the higher amount of bile was clogging our sterile filters.
Reviewer #2	
My question is: did the authors see any changes of membrane structures after bile and MBCD treatment? Did the investigators try to neutralize the effects of bile and MBCD by adding excess cholesterol from outside?	We addressed this question in the new SP Fig 1a. Our results indicate that MBCD and bile removed membrane lipid rafts as indicated by CTXB (SP Fig 1a). Exogenous cholesterol neutralized the effects of bile (SP Fig 1f-g) and MBCD (SP Fig 1b). Exogenous cholesterol blocked changes in membrane fluidity as indicated by NR12S (Sp Fig 1F) and the CWP1-NLuc induction that normally results from MBCD and bile treatments (SP Fig 1d-e).

In Fig.3, the use of membrane permeable cAMP (86P-cAMP) was used to assess the induction of encystation. Although the data is convincing, images shown in panel c are somewhat blurry, especially the DIC images. Authors might consider replacing the current image with improved one. Also, is it possible to show the images in higher magnification? Did authors try to assess the ESV biogenesis side by side with CWP1 expression? This should confirm the link between CWP expression and encystation as ESV biogenesis is the hallmark of cyst formation (encystation).	Regarding ESVs we took these images at relatively low mag (40x) to maximize quantification. Also the images are somewhat saturated because we set the exposure time to catch all cells expressing CWP1, as encystation progresses CWP1 levels rise and the brightest cells saturate our camera. We considered taking higher resolution images as suggested but unfortunately, we have depleted our supply of 8C6P-cAMP and despite multiple efforts to re-order, the only US supplier Axxora has not delivered (backordered since June 2021, last ordered Feb 3 2023). That said there should be no concern about the connection between CWP and ESV biogenesis. The uploaded PDF was relatively low resolution, we submitted full resolution Illustrator files for the re-submission but in case they are converted to low res we provide images to the right. If we zoom in on the presented image we can see ESVs. Images to the right are from the cAMP treatment. Second we took our experiments out to 48h and collected water resistant cysts. ESVs indicate cells in the process of encysting but a fully formed water resistant cyst wall is the ultimate hallmark of cyst formation. Since we were able to collect live water resistant cysts this indicates that all known and unknown processes necessary for encystation occurred. We assert that without proper ESV biogenesis the cyst wall would not have formed.	Fig.4. It appears that AC2 is involved in encystation and cyst production. However, the rationale for not participation of AC1 in encysting process requires additional discussion. Authors might consider showing AC1 localization and encystation (SP Fig.3) side by side with AC2 in Fig.4 (main text).	Point taken. We moved AC1-mNG localization so that it appears side by side with AC2-mNG (Fig 5a). Our focus was on the initiation of encystation and AC1 is only expressed later, but we see how a reader would still want to know about the role of AC1 since it gets induced during encystation. To address this point we did test if AC1 might also have a role in encystation. Knockdown indicates it has a role in cyst formation, but over expression does not induce encystation indicating it only functions at later steps of the process.	
Fig.5. The results on AC2 knockdown experiment impairs the encystation process is interesting. However, it will be more convincing if the authors show the	AC1 could regulate the same or different pathways, but we agree that its upregulation during encystation is intriguing. We now include functional studies of AC1. We screened for functional AC1 guide RNAs and found that	

results of AC1 knockdown and compare that with AC2 knockdown.	AC1 indeed has a role in regulating mid-late encystation. Knockdown reduces CWP1 expression at 24h of encystation (SP Fig 4c-e).
Fig.6. shows that the addition of cAMP or the increased level AC2 is sufficient for initiation of the encystation process. Is it possible then a change in the microenvironment of Giardia causing excess cAMP production should also trigger encystation? What would happen if AC1 produces excess cAMP? In addition, do the authors predict the presence of an encystation-related receptor in Giardia that is likely to modulate AC2 activation?	We cannot exclude the possibility that unknown changes in the microenvironment could initiate encystation. The Dawson lab showed that in mice encystation is induced in foci of infection. However, in our experience mice produce few cysts and cyst formation in foci does not reflect what has been reported in humans. Additionally, we cannot exclude the possibility that other cell stressors could induce cAMP signaling and encystation. Oxidative stress, ER stress, and simple starvation do not induce encystation. To our knowledge the only other cell stress reported to induced encystation is blocking sterol synthesis which is expected to impact cholesterol levels (PMID 17870055) We now include results for overexpression of AC1-mNG. Our data showed that AC1 is not sufficient to initiate encystation (SP Fig 4g-h). Moreover, we transformed AC2-g4159 into AC1-NLuc and our result showed that AC1 is not regulated by AC2 (SP Fig 7n). There may be one or more encystation-related receptors in Giardia. Cell density is known to impact encystation rates, the mechanism for detecting this is unknown. Additionally, there could be receptors or transceptors (transporter-receptors) that feed into the regulation of AC2. We are currently studying a candidate transceptor so stay tuned.
SP Fig.1 is informative. Authors might consider incorporating this figure of predicting cAMP signaling pathway in Giardia in the main text.	We made this figure 1 as suggested.
Reviewer #3	
>In rows 125 to 127 the authors state that the Small BiT (SmBiT) complementary peptide was fused with PKAc driven by its native promoter, and the Large BiT (LgBiT) was fused to PKAr driven by its native promoter. I understand that in this case, the proteins were not "endogenously tagged"? As in	All episomal cell lines are stable so long as we maintain antibiotic selection. The integrated cell lines are stable for several months, which is as long as anyone has cared to test. The SmBiT and LgBiT constructs are on episomal plasmids and have promoters driving their expression. The PKAc and PKAr-NanoLuc constructs are integrated into the genome using single site recombination where we linearize the plasmid inside the gene of interest

the case of PKAc-NanoLuc and PKAr-NanoLuc (lines 112) or with mNeonGreen and the Haloalkane dehalogenase (lines 117). What is this difference due to? It is a transient expression?	PMID: 21115739. These constructs typically do not include a promoter or start codon so that the only way to detect the tag is after successful integration into the endogenous site where just 1 of the 4 copies gets tagged. The choice of making G/PKA-NBit episomal was out of convenience for testing all the possible combinations of SmBit and LgBiT on the N and C terminal ends of PKAr and PKAc as is suggested in the NanoBiT user manual. This also allowed us to place the functional pair on a single plasmid.
>Figure 2d and e: DMSO control have no error bar. Do I have to interpret that it is very small and cannot be seen? The same does not happen in figure SP Figure 2b, where it can be seen that the result of the measurement of trophozoites that express GLPKA-NBit (Equivalent to the control of figure 2d and e) presents a dispersion of $\pm 8.3\%$. Even when the control is set to 100%, this 100% should represent the average of the samples and take into account the dispersion of the measurements. How many samples does each condition have? >The same observation can be made for all the figures with similar experiments.	Sp Figure 2b is a western blot so I think the reviewer may be referring to Sp Figure 2a where we measured cAMP levels during encystation. In this case the ELISA assay measures absolute concentration of cAMP compared to standards, so all time points have error bars. All of the NanoBit and NanoLuc data has been normalized to the control set to 100%. This is why the error bar for the control is non-existent. In Fig 2A of the original paper we normalized PKAr-Nluc and PKAc-Nluc to GAPDH-Nluc and there you can see some variability at time 0. The error bars for the experimental samples reflect standard deviation for our experiments. In most cases is necessary to normalize the data to a control because the level of luminescence changes rapidly after the addition of luminescent substrate. Comparing raw values would not be informative. We have updated the graphs to show the value of each independent biological experiment which typically averages 3 technical replicates so that variance is clearer.
Lines 149 to 151: At this point, I would say that the increase in membrane fluidity by treatment with MβCD or Bile triggers the elevation of intracellular cAMP and the activation of PKAc. I don't know if I would ascribe the entire effect to cholesterol depletion. The effect of treatment with Bile is more intense than that of MβCD and bile should be less specific in lipid uptake. MβCD isn't 100% specific to cholesterol, neither.	Agreed. The term membrane fluidity is now used throughout the paper but we do interpret this as a change in cholesterol due to previously published studies indicating the importance of cholesterol as well as our ability to block the induction of encystation by supplying exogenous cholesterol (see new SP Fig 1).
>The legends of the figures SP 3 and SP 4 are interchanged with each other.	Sorry for any confusion this caused, it is fixed.
>Line 211: The parentheses are not closed correctly	Fixed.
>Line 212 to 214: At what time after encystment was the expression of CWP2, CWP3, HCNCp, and the five GalNAc	16 h after encystation. The time points are now indicated in the figures to make the paper easier to follow..

measured? I cannot find this data in the text or in the legend of figure 5.	
>In figure 2d and 2e, the activation of PKA_Nbit is plotted as a decrease in luminescence, while in figure 4c a similar experiment is shown differently (% of dissociation). I believe that the way in which the results of similar experiments are presented should be consistent in order to avoid confusion.	Thank you for pointing this out, we have changed all of figures to % of association.
>The western blot of Supplementary Figure 3b is not representative of what is seen in Supplementary Figure 3c. I analyzed the wb with the program imagej, and the behavior presented in the bar graph does not match what is seen in the Western blot. How many images were analyzed for this result?	In response to this comment, we repeated our measurements of three independent biological replicates to make sure we did not have a mixup. Our re-measured values while not identical to our previous measurements (changing box size for ROI changes measured value) follow the same trend. We apologize for not presenting a blot that was more representative. We have replaced the blot with one that is.
>In Supplementary Figure 3a (sp fig. 3a), the amount of total cAMP is shown in pmol/ml. Is this the concentration of cAMP within the trophozoite or is this the concentration of the ELISA reaction mix? In materials and methods, it is specified that 20,000 cells per test were used, but in this figure it is marked as 2x10E6 cells (please check). I think the amount of cAMP should be expressed in pmol/20000 cells or similarly.	The cAMP measured is what was extracted from 2x10⁶ cells. We changed the Y axis to pmol/2x10⁶ cells and we corrected the material and methods.
> Lines 189 to 190: Are the same experiments of sp fig. 3a done with the clone AC2-g4159? Figure 4b shows only one point in time. Are these AC2-g4159 trophozoites in figure 4b incubated in normal medium or in encystation medium?	Sp Fig 3a is a western blot showing that exogenous cAMP upregulated encystation. We did not perform a western blot on AC2-g4159 after treatment with 8C6P-cAMP, but we do show that exogenous cAMP can rescue the encystation rate of AC2-g4159 by quantifying CWP1 positive cells in the current SP Figure 6e. The cAMP measurement for Figure 4b (now 5C) was performed after 1h of exposure to encystation medium since we wanted to examine the condition with maximal cAMP. The figure legend has been edited to make this clear.
>It took me a bit to understand the procedure for the construction of the Plasmids and the strategy in general. Would a slightly more detailed	We have all of plasmids labeled integrated or episomal in the primer table now. We also indicate the restriction site used for each integrated plasmid. The strategy for integration is straight forward see

description be possible with some diagram similar to what is shown in figure sp1? One of the most interesting aspects of this work was this, and the details of its realization are scarce.	https://www.ncbi.nlm.nih.gov/pmc/articles/PMC3019802/figure/F1/ it shows how we use single site recombination to endogenously tag genes. The figure is for a C-terminal 3HA tag, for this study we simply swapped out 3HA for mNeonGreen or NanoLuc and still use the same pKS vector. If we had included the correct Gourguechon reference we think the approach would have been clear from the start.
>Line 329: I can't find the restriction enzymes supposedly listed in Supplemental File 1.	We apologize for the omission; the supplemental file has been modified to include all restriction enzymes for ligation and linearization.
>Line 330 reference 38: I think this reference is more appropriate to explain the strategy used. https://journals.asm.org/doi/10.1128/EC.00190-10	You are correct, we apologize for accidentally selecting the wrong Gourguechon reference. This has been corrected.
After demonstrating that AC1 is not located in the plasma membrane and that it is preferentially expressed between 8 and 16 hours after the onset of encystment, AC1 is ignored by the authors. It is interesting to note that in the authors' words, G. lamblia is an organism with a cAMP machinery with minimal redundancies (Line 43), and while AC1 may not be playing a role in the encystment process, it is hard for me to ignore the fact that AC1 expression is being induced right during this process, as are MYB2, CWPs, and GalNAcs. Is it possible that AC1 expression is regulated by AC2? What would happen with the expression of AC1 in the clone AC2-g4159? Is it possible that AC1 expression is a necessary and independent step of AC2 to produce viable cysts?	We shared your curiosity about AC1 but initially planned to follow up on it later. In response to this comment, we developed a CRISPRi knockdown strain for AC1 (AC1-g300) and found that AC1-g300 knockdown downregulates CWP1 expression after 24h exposure to encystation medium (SP Fig 4c-e). This indicates that AC1 does have a role in promoting encystation. We also overexpressed AC1-mNG and found that AC1 is not sufficient to initiate encystation indicating it only functions at later steps which is consistent with its timing of expression. We also transformed AC2-g4159 into AC1-NLuc and our result showed that AC1 is not regulated by AC2 (SP Fig 7n), suggesting that its induction is independent of the initial cAMP pulse from AC2. Since cAMP alone is insufficient to promote fully formed cysts, we think this indicates additional signaling pathways are feeding into the regulation of encystation.

Reviewers' Comments:

Reviewer #1:

Remarks to the Author:

The paper has been revised according to the comments from reviewers and raised questions have been answered in a very nice way. I congratulate the authors for their work and suggest that this paper, that will be very important for the Giardia research community, is accepted.

Reviewer #2:

None

Reviewer #3:

Remarks to the Author:

The authors have conscientiously addressed all the queries raised. In the following sections, I listed my previous questions, presented the authors' responses, and conveyed my observations for each of them.

In regard to my observations, I believe that significant improvements have been made to the work. In this revised version, the authors have not only rectified previous errors but have also expanded their results by incorporating additional figures and increasing the amount of data presented within certain figures.

Question 1

>In rows 125 to 127 the authors state that the Small BiT (SmBiT) complementary pep de was fused with PKAc driven by its na ve promoter, and the Large BiT (LgBiT) was fused to PKAr driven by its na ve promoter. I understand that in this case, the proteins were not "endogenously tagged"? As in the case of PKAc-NanoLuc and PKAr- NanoLuc (lines 112) or with mNeonGreen and the Haloalkane dehalogenase (lines 117). What is this difference due to? It is a transient expression?

Author reply

All episomal cell lines are stable so long as we maintain antibiotic selection. The integrated cell lines are stable for several months, which is as long as anyone has cared to test. The SmBiT and LgBiT constructs are on episomal plasmids and have promoters driving their expression. The PKAc and PKAr-NanoLuc constructs are integrated into the genome using single site recombina on where we linearize the plasmid inside the gene of interest

PMID: 21115739. These constructs typically do not include a promoter or start codon so that the only way to detect the tag is after successful integration into the endogenous site where just 1 of the 4 copies gets tagged.

The choice of making GIPKA-NBit episomal was out of convenience for tes ng all the possible combina ons of SmBit and LgBiT on the N and C terminal ends of PKAr and PKAc as is suggested in the NanoBiT user manual. This also allowed us to place the functonal pair on a single plasmid.

My answer:

I am satisfied with the current response given by the author.

Question 2

>Figure 2d and e: DMSO control have no error bar. Do I have to interpret that it is very small and cannot be seen? The same does not happen in figure SP Figure 2b, where it can be seen that the result of the measurement of trophozoites that express GLPKA-NBit (Equivalent to the control of figure 2d and e) presents a dispersion of $\pm 8.3\%$. Even when the control is set to 100%, this 100% should represent the average of the samples and take into account the dispersion of the measurements. How many samples does each condition have?
>The same observation can be made for all the figures with similar experiments.

Author reply

Sp Figure 2b is a western blot so I think the reviewer may be referring to Sp Figure 2a where we measured cAMP levels during encystation. In this case the ELISA assay measures absolute concentration of cAMP compared to standards, so all the points have error bars. All of the NanoBit and NanoLuc data has been normalized to the control set to 100%. This is why the error bar for the control is non-existent. In Fig 2A of the original paper we normalized PKAr-Nluc and PKAc-Nluc to GAPDH-Nluc and there you can see some variability at 100. The error bars for the experimental samples reflect standard deviation for our experiments. In most cases it is necessary to normalize the data to a control because the level of luminescence changes rapidly after the addition of luminescent substrate. Comparing raw values would not be informative. We have updated the graphs to show the value of each independent biological experiment which typically averages 3 technical replicates so that variance is clearer.

My answer

My reference to figure Figure_2b was accurate in the original version. However, I have noticed that in the current version, this figure has been relocated to figure_2d.

Regarding the measurement of cAMP during encystment, it was presented in figure_4a in the previous version (now labeled as figure_3a in the current version).

These are just details, generally speaking, I am satisfied with the answer.

Question 3

Lines 149 to 151: At this point, I would say that the increase in membrane fluidity by treatment with M β CD or Bile triggers the elevation of intracellular cAMP and the activation of PKAc. I don't know if I would ascribe the entire effect to cholesterol depletion. The effect of treatment with Bile is more intense than that of M β CD and bile should be less specific in lipid uptake. M β CD isn't 100% specific to cholesterol, neither.

Author reply

Agreed. The term membrane fluidity is now used throughout the paper but we do interpret this as a change in cholesterol due to previously published studies indicating the importance of cholesterol as well as our ability to block the induction of encystation by supplying exogenous cholesterol (see new SP Fig 1).

My answer

I am satisfied with the changes made by the author.

Question 4

>The legends of the figures SP 3 and SP 4 are interchanged with each other.

Author reply

Sorry for any confusion this caused, it is fixed.

My answer

The changes have been implemented. Furthermore, the quantity of results displayed in figure SP4 has been significantly expanded.

Question 5

>Line 211: The parentheses are not closed correctly

Author reply

Fixed.

My answer

Ok

Question 6

>Line 212 to 214: At what time after encystment was the expression of CWP2, CWP3, HCNCP, and the five GalNAc measured? I cannot find this data in the text or in the legend of figure 5.

Author reply

16 h after encystment. The time points are now indicated in the figures to make the paper easier to follow.

My answer

Figure 6 is now much clearer and easier to understand. It's important to note that figure 6 in the current version corresponds to figure 5 in the previous version.

Question 7

>In figure 2d and 2e, the activation of PKA_Nbit is plotted as a decrease in luminescence, while in figure 4c a similar experiment is shown differently (% of dissociation). I believe that the way in which the results of similar experiments are presented should be consistent in order to avoid confusion.

Author reply

Thank you for pointing this out, we have changed all of figures to % of association.

My answer

Figures 3d and 3e are labeled as 'PKANBit Intensity (%)'.
Figure 5d is labeled as '% of association'.

Question 8

>The western blot of Supplementary Figure 3b is not representative of what is seen in Supplementary Figure 3c. I analyzed the wb with the program imagej, and the behavior presented in the bar graph does not match what is seen in the Western blot.
How many images were analyzed for this result?

Author reply

In response to this comment, we repeated our measurements of three independent biological replicates to make sure we did not have a mixup. Our re-measured values while not identical to our previous measurements (changing box size for ROI changes measured value) follow the same trend. We apologize for not presenting a blot that was more representative. We have replaced the blot with one that is.

My answer

The Western Blot (WB) in SP-figure 3b appears to better represent the results displayed in SP-figure 3c.

Question 9

>In Supplementary Figure 3a (sp fig. 3a), the amount of total cAMP is shown in pmol/ml. Is this the concentration of cAMP within the trophozoite or is this the concentration of the ELISA reaction mix? In materials and methods, it is specified that 20,000 cells per test were used, but in this figure it is marked as 2×10^6 cells (please check).
I think the amount of cAMP should be expressed in pmol/20000 cells or similarly.

Author reply

The cAMP measured is what was extracted from 2×10^6 cells. We changed the Y axis to pmol/ 2×10^6 cells and we corrected the material and methods.

My answer

Modifications have been applied to Figure 5c and the materials and methods section. However, it's worth noting that the y-axis of Figure SP-3a is still presented in pmol/ml.

Question 10

> Lines 189 to 190: Are the same experiments of sp fig. 3a done with the clone AC2-g4159? Figure 4b shows only one point in me. Are these AC2-g4159 trophozoites in figure 4b incubated in normal medium or in encystation medium?

Author reply

Sp Fig 3a is a western blot showing that exogenous cAMP upregulated encystation. We did not perform a western blot on AC2-g4159 after treatment with 8C6P-cAMP, but we do show that exogenous cAMP can rescue the encystation rate of AC2-g4159 by quantifying CWP1 positive cells in the current SP Figure 6e. The cAMP measurement for Figure 4b (now 5C) was performed after 1h of exposure to encystation medium since we wanted to examine the condition with maximal cAMP. The figure legend has been edited to make this clear.

My answer

I am satisfied with the changes made by the author.

Question 11

>It took me a bit to understand the procedure for the construction of the Plasmids and the strategy in general. Would a slightly more detailed description be possible with some diagram similar to what is shown in figure sp1? One of the most interesting aspects of this work was this, and the details of its realization are scarce.

Author reply

We have all of plasmids labeled integrated or episomal in the primer table now. We also indicate the restric on site used for each integrated plasmid. The strategy for integra on is straight forward see <https://www.ncbi.nlm.nih.gov/pmc/articles/PMC3019802/figure/F1/> it shows how we use single site recombina on to endogenously tag genes. The figure is for a C-terminal 3HA tag, for this study we simply swapped out 3HA for mNeonGreen or NanoLuc and s ll use the same pKS vector. If we had included the correct Gourguechon reference we think the approach would have been clear from the start.

My answer

I agree that a correct Gourguechon reference would have enhanced my comprehension of the author's methodology. I am satisfied with the implemented modifications.

Question 12

>Line 329: I can't find the restriction enzymes supposedly listed in Supplemental File 1.

Author reply

We apologize for the omission; the supplemental file has been modified to include all restriction enzymes for liga on and linearization.

My answer

The supplemental file has been updated and is now more comprehensible.

Question 13

>Line 330 reference 38: I think this reference is more appropriate to explain the strategy used. <https://journals.asm.org/doi/10.1128/EC.00190-10>

Author reply

You are correct, we apologize for accidentally selec ng the wrong Gourguechon reference. This has been corrected.

My answer

The Gourguechon reference has been successfully updated.

Question 14

After demonstrating that AC1 is not located in the plasma membrane and that it is preferentially

expressed between 8 and 16 hours after the onset of encystment, AC1 is ignored by the authors. It is interesting to note that in the authors' words, *G. lamblia* is an organism with a cAMP machinery with minimal redundancies (Line 43), and while AC1 may not be playing a role in the encystment process, it is hard for me to ignore the fact that AC1 expression is being induced right during this process, as are MYB2, CWPs, and GalNAcs.

Is it possible that AC1 expression is regulated by AC2? What would happen with the expression of AC1 in the clone AC2-g4159? Is it possible that AC1 expression is a necessary and independent step of AC2 to produce viable cysts?

Author reply

We shared your curiosity about AC1 but initially planned to follow up on it later. In response to this comment, we developed a CRISPRi knockdown strain for AC1 (AC1-g300) and found that AC1-g300 knockdown downregulates CWP1 expression after 24h exposure to encystment medium (SP Fig 4c-e). This indicates that AC1 does have a role in promoting encystment. We also overexpressed AC1-mNG and found that AC1 is not sufficient to initiate encystment indicating it only functions at later steps which is consistent with its timing of expression. We also transformed AC2-g4159 into AC1-NLuc and our result showed that AC1 is not regulated by AC2 (SP Fig 7n), suggesting that its induction is independent of the initial cAMP pulse from AC2. Since cAMP alone is insufficient to promote fully formed cysts, we think this indicates additional signaling pathways are feeding into the regulation of encystation.

My answer

I am satisfied with the answer provided by the author.

REVIEWERS' COMMENTS

Reviewer #1 (Remarks to the Author):

The paper has been revised according to the comments from reviewers and raised questions have been answered in a very nice way. I congratulate the authors for their work and suggest that this paper, that will be very important for the Giardia research community, is accepted.

We thank you for your efforts in reviewing our manuscript and helping us to improve it.

Reviewer #3 (Remarks to the Author):

The authors have conscientiously addressed all the queries raised. In the following sections, I listed my previous questions, presented the authors' responses, and conveyed my observations for each of them.

In regard to my observations, I believe that significant improvements have been made to the work. In this revised version, the authors have not only rectified previous errors but have also expanded their results by incorporating additional figures and increasing the amount of data presented within certain figures.

We thank you for your efforts in reviewing our manuscript and helping us to improve it. Due to the extensive reordering of figures we accidentally included some graphs that did not have the edited axis that were recommended. This was an oversight on our part but has been fixed in this revision. Specifically we changed labels in Figures 3d and 3e (Question 7) and Figure SP-3a (Question 9). The reviewer was satisfied by our responses to all other points.

Question 1

>In rows 125 to 127 the authors state that the Small BiT (SmBiT) complementary pep de was fused with PKAc driven by its na ve promoter, and the Large BiT (LgBiT) was fused to PKAr driven by its na ve promoter. I understand that in this case, the proteins were not "endogenously tagged"? As in the case of PKAc-NanoLuc and PKAr- NanoLuc (lines 112) or with mNeonGreen and the Haloalkane dehalogenase (lines 117). What is this difference due to? It is a transient expression?

Author reply

All episomal cell lines are stable so long as we maintain antibiotic selection. The integrated cell lines are stable for several months, which is as long as anyone has cared to test. The SmBiT and LgBiT constructs are on episomal plasmids and have promoters driving their expression. The PKAc and PKAr-NanoLuc constructs are integrated into the genome using single site recombina on where we linearize the plasmid inside the gene of interest

PMID: 21115739. These constructs typically do not include a promoter or start codon so that the only

way to detect the tag is after successful integration into the endogenous site where just 1 of the 4 copies gets tagged.

The choice of making GLPKA-NBit episomal was out of convenience for testing all the possible combinations of SmBit and LgBiT on the N and C terminal ends of PKAr and PKAc as is suggested in the NanoBiT user manual. This also allowed us to place the functional pair on a single plasmid.

My answer:

I am satisfied with the current response given by the author.

Question 2

>Figure 2d and e: DMSO control have no error bar. Do I have to interpret that it is very small and cannot be seen? The same does not happen in figure SP Figure 2b, where it can be seen that the result of the measurement of trophozoites that express GLPKA-NBit (Equivalent to the control of figure 2d and e) presents a dispersion of $\pm 8.3\%$. Even when the control is set to 100%, this 100% should represent the average of the samples and take into account the dispersion of the measurements. How many samples does each condition have?

>The same observation can be made for all the figures with similar experiments.

Author reply

Sp Figure 2b is a western blot so I think the reviewer may be referring to Sp Figure 2a where we measured cAMP levels during encystation. In this case the ELISA assay measures absolute concentration of cAMP compared to standards, so all time points have error bars. All of the NanoBit and NanoLuc data has been normalized to the control set to 100%. This is why the error bar for the control is non-existent. In Fig 2A of the original paper we normalized PKAr-Nluc and PKAc-Nluc to GAPDH-Nluc and there you can see some variability at time 0. The error bars for the experimental samples reflect standard deviation for our experiments. In most cases it is necessary to normalize the data to a control because the level of luminescence changes rapidly after the addition of luminescent substrate. Comparing raw values would not be informative. We have updated the graphs to show the value of each independent biological experiment which typically averages 3 technical replicates so that variance is clearer.

My answer

My reference to figure Figure_2b was accurate in the original version. However, I have noticed that in the current version, this figure has been relocated to figure_2d.

Regarding the measurement of cAMP during encystment, it was presented in figure_4a in the previous version (now labeled as figure_3a in the current version).

These are just details, generally speaking, I am satisfied with the answer.

Question 3

Lines 149 to 151: At this point, I would say that the increase in membrane fluidity by treatment with M β CD or Bile triggers the elevation of intracellular cAMP and the activation of PKAc. I don't know if I would ascribe the entire effect to cholesterol depletion. The effect of treatment with Bile is more intense than that of M β CD and bile should be less specific in lipid uptake. M β CD isn't 100% specific to cholesterol, neither.

Author reply

Agreed. The term membrane fluidity is now used throughout the paper but we do interpret this as a change in cholesterol due to previously published studies indicating the importance of cholesterol as well as our ability to block the induction of encystation by supplying exogenous cholesterol (see new SP Fig 1).

My answer

I am satisfied with the changes made by the author.

Question 4

>The legends of the figures SP 3 and SP 4 are interchanged with each other.

Author reply

Sorry for any confusion this caused, it is fixed.

My answer

The changes have been implemented. Furthermore, the quantity of results displayed in figure SP4 has been significantly expanded.

Question 5

>Line 211: The parentheses are not closed correctly

Author reply

Fixed.

My answer

Ok

Question 6

>Line 212 to 214: At what time after encystment was the expression of CWP2, CWP3, HCNCP, and the five GalNAc measured? I cannot find this data in the text or in the legend of figure 5.

Author reply

16 h after encystment. The time points are now indicated in the figures to make the paper easier to follow.

My answer

Figure 6 is now much clearer and easier to understand. It's important to note that figure 6 in the current version corresponds to figure 5 in the previous version.

Question 7

>In figure 2d and 2e, the activation of PKA_Nbit is plotted as a decrease in luminescence, while in figure 4c a similar experiment is shown differently (% of dissociation). I believe that the way in which the results of similar experiments are presented should be consistent in order to avoid confusion.

Author reply

Thank you for pointing this out, we have changed all of figures to **% of association**.

My answer

Figures 3d and 3e are labeled as 'PKANBit Intensity (%)'.

Figure 5d is labeled as '% of association'.

Response: The figure has been updated.

Question 8

>The western blot of Supplementary Figure 3b is not representative of what is seen in Supplementary Figure 3c. I analyzed the wb with the program imagej, and the behavior presented in the bar graph does not match what is seen in the Western blot.

How many images were analyzed for this result?

Author reply

In response to this comment, we repeated our measurements of three independent biological replicates to make sure we did not have a mixup. Our re-measured values while not identical to our previous measurements (changing box size for ROI changes measured value) follow the same trend. We apologize for not presenting a blot that was more representative. We have replaced the blot

with one that is.

My answer

The Western Blot (WB) in SP-figure 3b appears to better represent the results displayed in SP-figure 3c.

Question 9

>In Supplementary Figure 3a (sp fig. 3a), the amount of total cAMP is shown in pmol/ml. Is this the concentration of cAMP within the trophozoite or is this the concentration of the ELISA reaction mix? In materials and methods, it is specified that 20,000 cells per test were used, but in this figure it is marked as 2×10^6 cells (please check).

I think the amount of cAMP should be expressed in pmol/20000 cells or similarly.

Author reply

The cAMP measured is what was extracted from 2×10^6 cells. We changed the Y axis to pmol/ 2×10^6 cells and we corrected the material and methods.

My answer

Modifications have been applied to Figure 5c and the materials and methods section. However, it's worth noting that the **y-axis of Figure SP-3a is still presented in pmol/ml.**

Response: The figure has been updated.

Question 10

> Lines 189 to 190: Are the same experiments of sp fig. 3a done with the clone AC2-g4159? Figure 4b shows only one point in me. Are these AC2-g4159 trophozoites in figure 4b incubated in normal medium or in encystation medium?

Author reply

Sp Fig 3a is a western blot showing that exogenous cAMP upregulated encystation. We did not perform a western blot on AC2-g4159 after treatment with 8C6P-cAMP, but we do show that exogenous cAMP can rescue the encystation rate of AC2-g4159 by quantifying CWP1 positive cells in the current SP Figure 6e. The cAMP measurement for Figure 4b (now 5C) was performed after 1h of exposure to encystation medium since we wanted to examine the condition with maximal cAMP. The figure legend has been edited to make this clear.

My answer

I am satisfied with the changes made by the author.

Question 11

>It took me a bit to understand the procedure for the construction of the Plasmids and the strategy in general. Would a slightly more detailed description be possible with some diagram similar to what is shown in figure sp1? One of the most interesting aspects of this work was this, and the details of its realization are scarce.

Author reply

We have all of plasmids labeled integrated or episomal in the primer table now. We also indicate the restriction site used for each integrated plasmid. The strategy for integration is straight forward see <https://www.ncbi.nlm.nih.gov/pmc/articles/PMC3019802/figure/F1/> it shows how we use single site recombination to endogenously tag genes. The figure is for a C-terminal 3HA tag, for this study we simply swapped out 3HA for mNeonGreen or NanoLuc and still use the same pKS vector. If we had included the correct Gourguechon reference we think the approach would have been clear from the start.

My answer

I agree that a correct Gourguechon reference would have enhanced my comprehension of the author's methodology. I am satisfied with the implemented modifications.

Question 12

>Line 329: I can't find the restriction enzymes supposedly listed in Supplemental File 1.

Author reply

We apologize for the omission; the supplemental file has been modified to include all restriction enzymes for ligation and linearization.

My answer

The supplemental file has been updated and is now more comprehensible.

Question 13

>Line 330 reference 38: I think this reference is more appropriate to explain the strategy used.

<https://journals.asm.org/doi/10.1128/EC.00190-10>

Author reply

You are correct, we apologize for accidentally selecting the wrong Gourguechon reference. This has been corrected.

My answer

The Gourguechon reference has been successfully updated.

Question 14

After demonstrating that AC1 is not located in the plasma membrane and that it is preferentially expressed between 8 and 16 hours after the onset of encystment, AC1 is ignored by the authors. It is interesting to note that in the authors' words, *G. lamblia* is an organism with a cAMP machinery with minimal redundancies (Line 43), and while AC1 may not be playing a role in the encystment process, it is hard for me to ignore the fact that AC1 expression is being induced right during this process, as are MYB2, CWPs, and GalNAcs.

Is it possible that AC1 expression is regulated by AC2? What would happen with the expression of AC1 in the clone AC2-g4159? Is it possible that AC1 expression is a necessary and independent step of AC2 to produce viable cysts?

Author reply

We shared your curiosity about AC1 but initially planned to follow up on it later. In response to this comment, we developed a CRISPRi knockdown strain for AC1 (AC1-g300) and found that AC1-g300 knockdown downregulates CWP1 expression after 24h exposure to encystment medium (SP Fig 4c-e). This indicates that AC1 does have a role in promoting encystment. We also overexpressed AC1-mNG and found that AC1 is not sufficient to initiate encystment indicating it only functions at later steps which is consistent with its timing of expression. We also transformed AC2-g4159 into AC1-NLuc and our result showed that AC1 is not regulated by AC2 (SP Fig 7n), suggesting that its induction is independent of the initial cAMP pulse from AC2. Since cAMP alone is insufficient to promote fully formed cysts, we think this indicates additional signaling pathways are feeding into the regulation of encystation.

My answer

I am satisfied with the answer provided by the author.